# Revisiting Asymmetries in Black-box Link Stealing against Graph Neural Networks

**Paul Agbaje** [1]   **Habeeb Olufowobi** [1]

## Abstract

Graph Neural Networks (GNNs) are increasingly deployed on sensitive relational data, from social networks to healthcare records. However, their outputs can leak private graph structure, enabling link-stealing attacks that infer whether a connection between two entities existed in the training graph. While prior work demonstrates high average performance for such attacks, privacy is fundamentally a worst-case property, not an average one. The key question is whether an adversary can reliably compromise even a small set of critical links under strict precision constraints. We revisit posterior-only link-stealing attacks in a strict black-box setting and show that they remain effective at extremely low false-positive rates, revealing tail-risk vulnerabilities that current evaluations overlook. We further find that intra-class vulnerabilities are suppressed by geometric bottlenecks that collapse discriminative directions in posterior space. Building on this insight, we propose a geometry-aware reconditioning method that reshapes intra-class distances, substantially improving separability without harming reliability. Across multiple real-world graphs and GNNs, this diagnostic correction achieves up to $2\times$ higher success on intra-class pairs than generic attacks, redefining link-privacy evaluation as a tail-risk problem and revealing that posterior leakage remains substantially under-measured in current GNN deployments.

## 1. Introduction

Many real-world data sources, such as social networks, biological systems, and e-commerce platforms, are inherently non-Euclidean (Li et al., 2022; Bronstein et al., 2017). Unlike tabular or image data, they exhibit intricate topological structures that capture complex interactions among entities. Graphs provide a natural abstraction for these systems, with nodes representing entities and edges encoding relationships. For example, in social networks, users can be modeled as nodes and their friendships or interactions as edges (Wilson et al., 2009). The success of deep learning has extended into this setting through Graph Neural Networks (GNNs), which generalize neural architectures to graph-structured data. By propagating and aggregating information across neighborhoods, GNNs learn representations that enable strong performance on tasks such as semi-supervised node classification, link prediction, and graph classification (Gasteiger et al., 2018; Zhang & Chen, 2018; Agbaje et al., 2026b).

As GNNs become widely deployed, they expose new attack surfaces. The representations and predictions they generate can inadvertently encode sensitive relational information that adversaries may exploit. For example, graph embeddings hosted on cloud platforms can be reverse-engineered to uncover private attributes or hidden structural patterns of the underlying graph (Zhang et al., 2022). Furthermore, trained GNNs constitute valuable intellectual property, as organizations invest substantial effort in curating large graph datasets and training robust models, making them attractive targets. Understanding how link information can be inferred from deployed GNNs is critical for strengthening trust and privacy in online ecosystems.

Recent work has shown that posterior-based link-stealing attacks on GNNs are feasible by exploiting the tendency of message passing to induce higher posterior similarity between connected nodes. He et al. (He et al., 2021a) demonstrated that simple similarity-based attacks succeed under minimal assumptions, while Zhang et al. (Zhang et al., 2023) identified a strong asymmetry in vulnerability, with inter-class links being substantially easier to infer than intra-class ones. However, existing evaluations obscure practical privacy risk by focusing on average performance across all node pairs. As privacy is a worst-case property (Carlini et al., 2022), an adversary benefits most from reliably inferring a small subset of sensitive links at low false positive rates. Moreover, current threshold-based attacks exhibit systematic blind spots: even with class-conditional stratifi-

[1]Department of Computer Science and Engineering, University of Texas at Arlington, TX, USA. Correspondence to: Paul Agbaje <poa7959@mavs.uta.edu>, Habeeb Olufowobi <habeeb.olufowobi@uta.edu>.

*Proceedings of the 43rd International Conference on Machine Learning*, Seoul, South Korea. PMLR 306, 2026. Copyright 2026 by the author(s).

cation, intra-class links remain difficult to exploit, indicating a deeper geometric collapse in posterior space that limits discriminative power.

Understanding these bottlenecks is essential for a more complete assessment of privacy risks in GNNs. We therefore pose three research questions: (**RQ-1**) *Do posterior-based link-stealing attacks remain reliable in the tail of the confidence distribution, where adversaries must operate at low false positive rates?* (**RQ-2**) *If tail reliability degrades, what mechanism explains why intra-class links are systematically harder to exploit than inter-class ones?* (**RQ-3**) *Given this disparity, can intra-class leakage be deliberately amplified without sacrificing overall inference reliability?* These questions are fundamentally non-trivial because link privacy departs from membership inference in i.i.d. settings: leakage is not driven by sample-level overfitting, but by the joint geometry of node-pair posteriors shaped by graph structure. We identify a concrete geometric failure mode in which posterior covariance for intra-class pairs collapses along low-variance directions, suppressing discriminative signal and obscuring true risk. Correcting this collapse reveals that existing evaluations materially underestimate privacy leakage in GNNs.

In this work, we reframe link privacy as a tail-risk problem and introduce a geometry-aware correction that improves intra-class separability. By shifting evaluation from average-case metrics to worst-case reliability under strict false-positive constraints, we show that latent vulnerabilities persist even in seemingly secure regimes. In summary, we make the following contributions:

- We provide the first systematic evaluation of posterior-based link-stealing attacks under extremely low false positive rates, showing that average-case metrics substantially understate privacy risks.
- We identify geometric bottlenecks that suppress intra-class vulnerabilities, revealing how posterior covariance structures collapse within predicted classes.
- We propose per-class covariance whitening, which amplifies low-variance class-specific directions and substantially improves intra-class separability.
- Experiments on multiple real-world graphs and GNN architectures show consistent improvements in both average performance and the low false-positive regime.

## 2. Related Work

**MIAs on GNNs.** Prior work has shown that GNNs leak membership information under adversarial analysis (Agbaje et al., 2026a). Duddu et al. (Duddu et al., 2020) demonstrated that node embeddings can be exploited to distinguish training from non-training nodes, while Olatunji et al. (Olatunji et al., 2021) showed that structural signals

further amplify this leakage. He et al. (He et al., 2021b) established that node-level MIAs remain effective even with limited adversarial knowledge. While node-level MIAs target individual training participation, edge-level MIAs instead infer the existence of relationships between nodes, raising distinct risks to relational privacy.

**Link-level MIAs.** At the edge level, He et al. (He et al., 2021a) introduced a link-stealing attack that infers training edges from posterior similarities induced by message passing under a black-box threat model. Zhang et al. (Zhang et al., 2023) refined this setting by showing that inter- and intra-class edges exhibit systematically different exposure levels, enabling conditional thresholding to improve attack performance. Wu et al. (Wu et al., 2024b) extended link inference by incorporating node attributes and graph features within a shadow-model framework. In this work, we focus on the most realistic threat model introduced by He et al. (He et al., 2021a) and refined by Zhang et al. (Zhang et al., 2023), where the adversary has only black-box access to the victim GNN and relies exclusively on posterior outputs. We further analyze the fundamental limitations and performance bottlenecks of such posterior-only attacks, and investigate whether identifying these bottlenecks enables more effective link inference.

## 3. Preliminaries

### 3.1. Target Graph Neural Network

An attributed graph is denoted by $\mathcal{G} = (\mathcal{V}, \mathcal{E}, \mathcal{X})$, where $\mathcal{V}$ is the set of nodes, $\mathcal{E} \subseteq \mathcal{V} \times \mathcal{V}$ is the edge set, and $\mathcal{X} \in \mathbb{R}^{|\mathcal{V}| \times n}$ is the feature matrix, with $\mathcal{X}_u \in \mathbb{R}^n$ representing the $n$-dimensional attributes of node $u$. The topology can also be written as an adjacency matrix $\mathcal{A} \in \{0, 1\}^{|\mathcal{V}| \times |\mathcal{V}|}$, where $\mathcal{A}_{uv} = 1$ if an edge connects $u$ and $v$, and 0 otherwise. A GNN maps $\mathcal{G}$ to a matrix of node representations $\mathcal{Z} \in \mathbb{R}^{|\mathcal{V}| \times h}$, with $\mathcal{Z}_u$ denoting the $h$-dimensional embedding of node $u$. These embeddings provide a foundation for tasks such as node classification, link prediction, and whole-graph classification. Details of the GNN training and prediction procedures are provided in Appendix A.

### 3.2. Link-Stealing Attack

A link-stealing attack aims to determine whether a pair of nodes in the training graph of a target model are connected, i.e., to infer the value of $\mathcal{A}_{uv}$ for nodes $u, v \in \mathcal{V}$. Successful inference enables an adversary to reconstruct portions of the underlying training graph $\mathcal{G}$, raising significant privacy concerns (He et al., 2021a). GNNs update node representations by aggregating information from neighbors; hence, embeddings of adjacent nodes tend to be more similar than those of unrelated nodes. An attacker can exploit this property by comparing the posteriors of the target model for two nodes.

Let $p(u)$ and $p(v)$ denote the posterior probability vectors of nodes $u$ and $v$, respectively. To quantify their similarity, we apply a distance function $d\big(p(u), p(v)\big)$. We define a link score as: $s(u, v) = 1 - d\big(p(u), p(v)\big)$, so that larger values of $s(u, v)$ suggest stronger evidence of an edge. Formally, the attack can be cast as a binary hypothesis test:

$$H_0 : (u, v) \notin \mathcal{E} \quad (\mathcal{A}_{uv} = 0),$$
$$H_1 : (u, v) \in \mathcal{E} \quad (\mathcal{A}_{uv} = 1). \tag{1}$$

Given a decision threshold $\gamma$, the adversary predicts the presence or absence of an edge as follows:

$$\mathcal{C}\big(f_{\text{tar}}, (u, v)\big) = \begin{cases} H_1, & s(u, v) \geq \gamma, \\ H_0, & s(u, v) < \gamma, \end{cases} \tag{2}$$

where $f_{\text{tar}}$ denotes the victim model. The choice of $\gamma$ controls the trade-off between true and false positives.

### 3.3. Threat Model

**Adversary's Objective.** The adversary's goal is to determine whether a pair of nodes $(u, v)$ in the training graph $\mathcal{G}$ used to train a target GNN are connected—constituting an edge-level MIA that threatens the confidentiality of relationships in the underlying data and may reveal sensitive information held by the model owner.

**Attacker's Capability.** We assume that the attacker has *black-box* access to the target model $f_{\text{tar}}$: the attacker can submit queries and observe the output posteriors $p(u)$ for individual nodes $u$, but has no access to the model's parameters, architecture, or training procedure. Each posterior $p(u)$ is a probability vector over the label space, representing the model's prediction for node $u$. This limited interface reflects a practical scenario in which GNNs are deployed through Machine-Learning-as-a-Service (MLaaS) platforms (e.g., AWS with DGL support (Adeshina, 2020) and Microsoft Azure's Spectra (Balakreshnan, 2021)), where clients can access predictions via an API but cannot inspect the internal model (Wu et al., 2024a). Such a setting captures a strong, yet realistic, threat model for privacy analysis of GNNs (He et al., 2021a; Zhang et al., 2023).

## 4. Evaluating Link–Stealing Attacks

### 4.1. Tail Reliability of Link-Stealing Attacks

To answer **RQ-1**, we benchmark two posterior-only attacks: (i) the generic posterior–distance attack of He et al. (He et al., 2021a) and (ii) the inter-/intra-class stratification of Zhang et al. (Zhang et al., 2023), which exploits uneven vulnerability across semantic groups. We evaluate AUC and tail reliability TPR@$10^{-3}$ FPR on standard citation benchmarks (Cora, Citeseer, Pubmed) using a GCN model; results are reported in Table 1.

**Confidence-conditioned grouping (CC-LS).** To probe the attack behavior without relying on predicted labels, we introduce a confidence-conditioned grouping that stratifies solely on posterior confidence. For each node posterior $p$, compute the margin $m(p) = p_{(1)} - p_{(2)}$. For a candidate pair $(u, v)$ define the pair confidence

$$M(u, v) = \min\{m(p_u), m(p_v)\}.$$

The values $M(u, v)$ are partitioned into $K$ disjoint bins $\{g_0, \ldots, g_{K-1}\}$ using empirical quantiles of the distribution. Each pair $(u, v)$ is assigned to exactly one bin $g_k$. Pairs are scored over the same set of distances in the generic link stealing attack (He et al., 2021a). Within each bin $g$, a two-component Gaussian mixture is fitted to the scores $\{s(u, v) : (u, v) \in g\}$ to separate putative edges from non-edges. Conditioning on $M(u, v)$ yields an attacker-centric grouping mechanism distinct from class-based stratification. Rather than relying on predicted labels, it leverages posterior confidence as a lens to probe which subsets of node pairs are most reliably exploitable.

**Observation 1.** We observe that posterior-based attacks remain effective even at extremely low FPR. As shown in Fig. 1(a) and Table 1, at a false positive rate of only 0.1%, a generic stealing attack on Cora with correlation distance achieves a TPR of 20.3% with an average-case AUC of 92.6%. In practical terms, with at most one false alarm per $1,000$ pairs, an attacker can still recover roughly one in five true edges—a nontrivial exposure for sensitive datasets. These results underscore the importance of evaluating ROC tails rather than relying solely on aggregate measures.

**Observation 2.** The relation between posterior confidence margin and attack behavior varies across groups and datasets. Let $m(p) = p_{(1)} - p_{(2)}$ denote a node's margin between the top two posterior coordinates. For inter-class pairs, lower margins often improve both AUC and TPR at very small FPR: uncertainty spreads posterior mass into secondary coordinates and increases directional contrast between classes. This effect is strong on Cora and PubMed but weaker on Citeseer (see Appendix B, Table 3) where the non-link distance distribution has lower density across most ranges—few non-links achieve large separation from links—so the benefit of margin conditioning is reduced. For intra-class pairs, we observe a consistent, operating point–dependent pattern across all datasets: low margins yield higher AUC because larger but noisier tails help ranking, whereas high margins give better TPR at low FPR since cleaner tails reduce extreme overlap. For instance, on Cora, intra-class TPR rises from 0.120 at bin $g_0$ to 0.312 at bin $g_1$, while AUC drops from 0.679 to 0.666 using cosine distance. These observations indicate that inter and intra-class links occupy different regions of posterior space. Hence, distances for intra-class links should be computed such that the peculiarities of the different regions are accounted for.

*Table 1.* AUC and TPR@0.1% for link–nonlink discrimination on Cora using eight distance metrics. Rows separate generic distances, inter-class and intra-class subsets, and CC-LS at two operating points $g_0$ (low margin) and $g_1$ (high margin). Bold entries denote the best score per block. Results highlight how confidence margin and class relation (inter vs. intra) interact with distance choice.

| | | cosine | | euclidean | | sqeuclidean | | correlation | | cityblock | | chebyshev | | braycurtis | | canberra | |
|---|---|---|---|---|---|---|---|---|---|---|---|---|---|---|---|---|---|
| | | AUC | TPR | AUC | TPR | AUC | TPR | AUC | TPR | AUC | TPR | AUC | TPR | AUC | TPR | AUC | TPR |
| | Generic | 0.914 | 0.18 | 0.875 | 0.165 | 0.875 | 0.165 | 0.926 | 0.203 | 0.879 | 0.131 | 0.852 | 0.129 | 0.879 | 0.131 | 0.721 | 0.098 |
| | Inter | **0.895** | **0.246** | **0.848** | **0.197** | **0.848** | **0.197** | **0.923** | 0.164 | **0.863** | **0.262** | **0.833** | **0.197** | **0.863** | **0.262** | **0.771** | **0.131** |
| | Intra | 0.699 | 0.203 | 0.631 | 0.184 | 0.630 | 0.184 | 0.746 | **0.229** | 0.640 | 0.146 | 0.603 | 0.143 | 0.640 | 0.146 | 0.554 | 0.109 |
| | CC-LS ($g_0$) | 0.892 | 0.080 | 0.833 | 0.111 | 0.833 | 0.111 | 0.907 | 0.084 | 0.842 | 0.080 | 0.806 | 0.093 | 0.842 | 0.080 | 0.738 | 0.089 |
| Cora | CC-LS ($g_1$) | **0.921** | **0.281** | **0.899** | **0.204** | **0.899** | **0.204** | **0.944** | **0.323** | **0.901** | **0.198** | **0.887** | **0.158** | **0.901** | **0.198** | 0.715 | **0.135** |
| | Inter—CC-LS ($g_0$) | **0.915** | **0.300** | **0.874** | **0.240** | **0.874** | **0.240** | **0.924** | 0.200 | **0.891** | **0.320** | **0.842** | **0.240** | **0.891** | **0.320** | **0.819** | **0.160** |
| | Inter—CC-LS ($g_1$) | 0.782 | 0.091 | 0.635 | 0.000 | 0.635 | 0.000 | 0.864 | 0.091 | 0.658 | 0.000 | 0.627 | 0.000 | 0.657 | 0.000 | 0.550 | 0.091 |
| | Intra—CC-LS ($g_0$) | **0.679** | 0.120 | **0.644** | 0.139 | **0.644** | 0.139 | 0.717 | 0.074 | 0.649 | 0.106 | **0.622** | 0.120 | 0.649 | 0.106 | **0.592** | 0.116 |
| | Intra—CC-LS ($g_1$) | 0.666 | **0.314** | 0.575 | **0.223** | 0.573 | **0.223** | **0.779** | **0.363** | 0.585 | **0.215** | 0.539 | **0.167** | 0.585 | **0.215** | 0.519 | **0.139** |

### 4.2. Posterior-Space Separability

We begin addressing **RQ-2** by formalizing the distance score used throughout our analysis. Given a candidate pair $(u, v)$, its distance score is $S(u,v) = d(p^{(u)}, p^{(v)})$, where $p^{(u)}, p^{(v)} \in \Delta^{\Lambda-1}$ denote the model posteriors. For a node $u$, we write $p^{(u)} \in \Delta^{\Lambda-1}$ for its posterior vector and $\hat{y}(u) = \arg\max_i p_i^{(u)}$ for its predicted label. We define the grouping map $Q(u) = \hat{y}(u)$. For a node pair $(u, v)$:

$$\begin{aligned} \mathcal{R} &= \{(u,v) : Q(u) = Q(v)\} \quad \text{(intra)} \\ \mathcal{I} &= \{(u,v) : Q(u) \neq Q(v)\} \quad \text{(inter)} \end{aligned} \tag{3}$$

Let $Y_{uv} \in \{0, 1\}$ denote the true link label ($Y_{uv} = 1$ for an edge, $Y_{uv} = 0$ otherwise). Within any fixed stratum $\mathsf{G} \in \{\mathcal{R}, \mathcal{I}\}$, we consider the one-dimensional score distributions $f_1 = \mathcal{L}\big(S(u,v) \big| Y_{uv} = 1, (u,v) \in \mathsf{G}\big)$ and $f_0 = \mathcal{L}\big(S(u,v) \big| Y_{uv} = 0, (u,v) \in \mathsf{G}\big)$. With equal priors, the Bayes error for deciding $Y$ from $S$ equals $P_{e,\min} = \frac{1}{2}\big(1 - \mathrm{TV}(f_1, f_0)\big)$, where TV is the total variation distance. For $p \in \Delta^{\Lambda-1}$, let $m(p) = p_{(1)} - p_{(2)}$ denote the top-1 margin. We compute the scores $d \in \mathcal{D}$ with a set of distances:

$$\mathcal{D} = \{\|p - q\|_1, \|p - q\|_2, \|p - q\|_2^2, \|p - q\|_\infty\}.$$

**Definition 4.1.** A pair $(p, q)$ is *intra* if $\arg\max_i p_i = \arg\max_i q_i$, and *inter* otherwise.

**Lemma 4.2.** *Let* $a = \arg\max_i p_i$ *and* $b \neq a$. *Then* $p_a - p_b \geq m(p)$.

*Proof.* By definition $p_a = p_{(1)}$ is the largest coordinate and $p_b \leq p_{(2)}$ is no larger than the second largest. Hence $p_a - p_b \geq p_{(1)} - p_{(2)} = m(p)$. □

**Lemma 4.3.** *Let* $\Delta_a = p_a - q_a$ *and* $\Delta_b = p_b - q_b$ *for indices* $a, b$. *Then*

$$\begin{aligned} \|p - q\|_1 &\geq |\Delta_a| + |\Delta_b| \geq |\Delta_a - \Delta_b|, \\ \|p - q\|_2^2 &\geq \Delta_a^2 + \Delta_b^2 \geq \tfrac{1}{2}(\Delta_a - \Delta_b)^2. \end{aligned}$$

*Proof.* The first inequality holds since $\|p - q\|_1$ is the sum over all coordinates. The inequality $|\Delta_a| + |\Delta_b| \geq |\Delta_a -$

$\Delta_b|$ follows from the triangle inequality $|x| + |y| \geq |x - y|$. For $\ell_2$, since $\|p - q\|_2^2$ sums all squared differences, it is at least $\Delta_a^2 + \Delta_b^2$. Finally, by Cauchy–Schwarz, $x^2 + y^2 \geq \frac{1}{2}(x - y)^2$, which yields the last bound. □

**Theorem 4.4.** *Fix any* $\varepsilon > 0$ *and any* $d \in \mathcal{D}$. *There exist intra-pairs with labels* $Y = 1$ *and* $Y = 0$ *whose scores differ by less than* $\varepsilon$. *Thus, distance-based scoring cannot guarantee separation of links and non-links within* $\mathcal{R}$.

*Proof.* Pick a class $a$ and a sufficiently small $\delta > 0$. Let $p = (1 - \delta)e_a + \delta r$ and $q = (1 - \delta)e_a + \delta r'$ with arbitrary $r, r' \in \Delta^{\Lambda-1}$. Then $\arg\max p = \arg\max q = a$ (intra) and, by continuity on the simplex, $\|p - q\|_1 \leq 2\delta$, $\|p - q\|_2 \leq \sqrt{2}\,\delta$, $\|p - q\|_\infty \leq \delta$. Consider two such intra-pairs (one with $Y = 1$, one with $Y = 0$) with the same $\delta$. Choosing $\delta$ sufficiently small enforces score proximity $< \varepsilon$. □

*Remark* 4.5. For intra-class pairs, both posteriors peak on the same coordinate. This alignment cancels the dominant entries, driving distances toward zero regardless of whether an edge exists. As a result, discrimination relies only on weak variations in the tail coordinates, producing heavy overlap between link and non-link distributions. This formalizes the intuition that intra-class pairs create a structural bottleneck: attack success is suppressed because informative differences are easily masked.

**Theorem 4.6.** *Assume* $m(p) \geq m_0 > 0$ *and* $m(q) \geq m_0$. *If* $(p, q)$ *is inter with* $a = \arg\max p$ *and* $b = \arg\max q$ *($a \neq b$), let* $\mathcal{M} := m(p) + m(q)$. *Then*

$$\|p - q\|_1 \geq \mathcal{M} \geq 2m_0, \quad \|p - q\|_2 \geq \tfrac{\mathcal{M}}{\sqrt{2}} \geq \sqrt{2}\,m_0,$$

$$\|p - q\|_\infty \geq \tfrac{\mathcal{M}}{2} \geq m_0, \quad \|p - q\|_2^2 \geq \tfrac{1}{2}\mathcal{M}^2 \geq 2m_0^2.$$

*Proof.* Let $a = \arg\max p$ and $b = \arg\max q$ with $a \neq b$. By Lemma 4.2, $p_a - p_b \geq m(p)$ and $q_b - q_a \geq m(q)$. Hence

$$\begin{aligned} (\Delta_a - \Delta_b) &= (p_a - q_a) - (p_b - q_b) \\ &= (p_a - p_b) - (q_a - q_b) \geq m(p) + m(q). \end{aligned} \tag{4}$$

Apply Lemma 4.3 to obtain the $\ell_1$ and $\ell_2$ bounds; the $\ell_\infty$ bound follows from $\|p - q\|_\infty \geq \frac{1}{2}(|\Delta_a| + |\Delta_b|) \geq \frac{1}{2}|\Delta_a - \Delta_b|$. Squaring the $\ell_2$ bound gives the stated squared-Euclidean inequality. □

*Remark* 4.7. For inter-class pairs, the two posteriors concentrate on different dominant coordinates. This structural misalignment enforces a persistent gap: the winning class of $p$ is suppressed in $q$, and vice versa. Thus, distances do not easily collapse to zero, even if the remaining coordinates are nearly identical. This explains why inter-class pairs consistently exhibit clearer separation between link and non-link distributions: their score distributions are shifted apart by a deterministic margin floor. In contrast, Theorem 4.4 shows that intra-class pairs lack such a floor, making them more vulnerable to overlap.

### 4.3. Investigating Attack Bottlenecks for Intra-class Links

To address **RQ-2**, we empirically examine why intra-class links remain substantially harder to exploit than inter-class pairs. Following Carlini et al. (Carlini et al., 2022), we evaluate attack performance in the low–FPR regime using ROC curves with TPR plotted against FPR on a logarithmic scale. Results for the Cora dataset are shown in Fig. 1 (a), while corresponding results for Citeseer and PubMed are deferred to Appendix C, Fig. 6 (a,e). Complementary visualizations, including score distributions and box plots, are provided for Cora in Fig. 1(b–d) and for Citeseer and PubMed in Appendix C, Fig. 6 (b–d,f–h), revealing systematic differences in score separability between inter- and intra-class pairs.

A key observation is that the baseline distance metrics used in prior work, such as cosine or Euclidean distance, are *geometry-driven measures* that do not account for the probabilistic distribution nature of posteriors. To test this effect, we evaluate probability-aware divergences specifically designed for comparing distributions. Specifically, we consider the Bhattacharyya and Fisher–Rao distances, both of which capture distributional similarity in a way that considers the underlying probabilistic structure (Bhattacharyya, 1943; Rao et al., 1945). Results for Cora dataset are shown in Fig. 2.

**Observation 3.** Across datasets, we observe that distance scores for intra-class pairs are centered near zero, whereas inter-class pairs exhibit substantially higher mean distances. This pattern aligns with Theorem 4.4 and Theorem 4.6: intra-class pairs collapse their distances due to shared argmax entries, while inter-class pairs are separated by a deterministic margin floor. Empirically, this means intra-class distances are dominated by the largest posterior coordinates, shrinking the dynamic range and impairing separability between links and non-links. Consistent with this theoretical effect, our results show that inter-class pairs achieve clear separa-

tion under the same metrics, while intra-class pairs exhibit significant overlap. Thus, the intra-class bottleneck arises because geometric distances compress posterior information precisely in the region where discrimination is most needed.

**Observation 4.** This bottleneck persists even when geometric distances are replaced with probability-aware divergences (Fig. 2). For example, the Bhattacharyya distance collapses intra-class scores toward zero, reproducing the same degeneracy predicted by Theorem 4.4. The Fisher–Rao distance provides a broader spread, but fails to produce clear link–non-link separation: both distributions continue to overlap heavily. Thus, while probability-respecting divergences adjust scale, they fail to resolve the core limitation: intra-class pairs lack discriminative variation within the dominant posterior mode.

These observations highlight a basic limitation of existing metrics and underscore the need for approaches that recover discriminative structure within predicted classes.

## 5. Link Stealing for Intra-Class Pairs

### 5.1. Variance–Discriminability Misalignment

We argue that the collapse of intra-class separability arises because, in some cases, variance and discriminability are misaligned, and because anisotropy differs across predicted classes. To examine this, we first decompose the covariance of intra-class posterior differences. For each predicted class, we project differences onto the eigenvectors and compare two cumulative curves: the fraction of variance explained by the top $m$ components, and the fraction of discriminability (measured via Fisher ratio between link and non-link pairs) explained by the same components. As shown in Fig. 3(a) for the GAT/Cora setting, the top two eigenmodes almost entirely capture variance, yet discriminability accumulates more slowly and requires four to six modes. This analysis shows that intra-class collapse stems from a misalignment between variance and discriminability. A small number of high-variance directions dominate the geometry of posterior differences, yet these directions contribute little to distinguishing links from non-links. Instead, discriminative structure is spread across other modes whose variance is modest but whose separation power is critical. Conventional distances ignore this mismatch, collapsing intra-class pairs toward zero and obscuring discriminative signals.

Next, to test whether this anisotropy is global or class-specific, we compare the dominant subspaces across predicted classes by computing principal angles between their top eigenvectors. The heatmaps in Fig. 3(b) show substantial angles in the range of 0.5–1.0 radians (approximately $30°$–$60°$) between classes, indicating that each class has its own anisotropy structure. We confirm this with a cross-whitening test by fitting a whitening transform from one

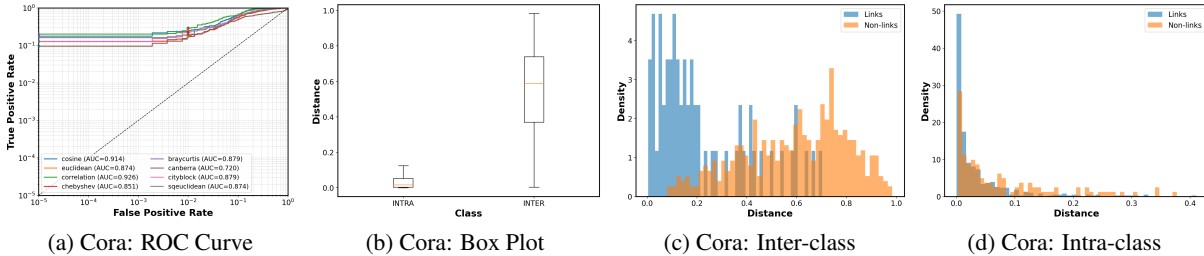

(a) Cora: ROC Curve   (b) Cora: Box Plot   (c) Cora: Inter-class   (d) Cora: Intra-class

*Figure 1.* Distance-based link prediction on the Cora citation dataset. We report ROC curves across distance metrics, together with box plots and cosine-distance distributions for inter-class and intra-class pairs.

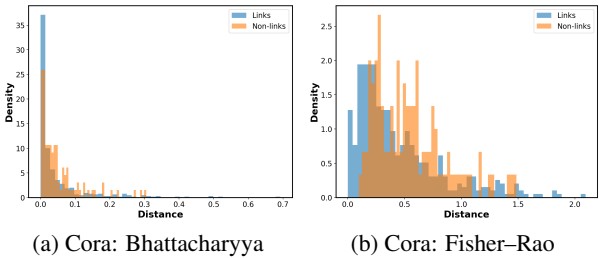

(a) Cora: Bhattacharyya   (b) Cora: Fisher–Rao

*Figure 2.* Distributions of intra-class link and non-link pairs on the Cora dataset using two probability divergences. (a) Bhattacharyya distance and (b) Fisher–Rao distance both reveal limited separability within intra-class pairs.

class's covariance and applying it to the intra-class pairs of another class. As shown in Fig. 3(c), whitening enhances separability along the diagonal (within-class pairs), but degrades performance on many off-diagonals. This indicates that anisotropy varies across classes and that a single global transformation cannot adequately address class-specific distortions. This motivates our attack strategy to restore intra-class separability without degrading inter-class structure, addressing **RQ-3** and revealing that attackers can amplify vulnerabilities without reducing overall reliability.

### 5.2. Geometry-aware Reconditioning

Empirically, intra-posterior differences collapse because there are cases where variation is dominated by a few high-variance but weakly discriminative directions. Conventional distances may not fully consider the directions that actually carry a discriminative signal. The consequence is that scores for intra-class pairs are compressed toward zero, masking the structure needed to distinguish links from non-links. To correct this distortion, we propose a Per-Class Whitening (PCW) transformation that reconditions the geometry of posteriors to recover suppressed discriminative structure. While global whitening or zero-phase component analysis (Wadia et al., 2020) removes dominant variance directions, such methods assume a shared anisotropy across all classes. Our empirical analysis (Fig. 3) shows otherwise: each predicted class induces a distinct anisotropy pattern, with class-specific subspaces where discriminative information is concentrated. A global transform would blur these

differences and potentially distort inter-class geometry.

Accordingly, for each predicted class $c$, we fit a linear transformation $W_c$ in a variance-stabilized power space $x = \phi_\tau(p) = p^\tau$ (with $\tau = \frac{1}{2}$ by default). We estimate the class-conditional covariance $\widehat{\Sigma}_c$, perform an eigendecomposition, and construct a whitening operator:

$$W_c = U_c \operatorname{diag}(\tilde{\gamma}_1, \ldots, \tilde{\gamma}_K) U_c^\top,$$

where $\tilde{\gamma}_j$ are inverse square-root eigenvalues. Each intra-class endpoint is then reconditioned as

$$z_j = W_c\big(\phi_\tau(p_j) - \mu_c\big), \quad j \in \{0, 1\},$$

before distance computation. This transformation amplifies low-variance, high-discriminability components and suppresses spurious variance directions, restoring separability between link and non-link distributions that were previously collapsed. Since anisotropy differs across classes, we fit one $W_c$ per predicted class and apply it only when both endpoints belong to that class; inter-class pairs, which already exhibit strong natural separation, are left unaltered.

## 6. Experiments

### 6.1. Experimental Setup.

**Datasets.** We evaluate on six real-world benchmarks: Cora, Citeseer, PubMed (Sen et al., 2008; Namata et al., 2012), AIDS (Bunke & Riesen, 2008), Amazon Photos, and CS (Shchur et al., 2018). Cora, Citeseer, and PubMed are citation networks with documents as nodes and citation links as edges. AIDS is a molecular graph dataset in which nodes represent atoms and edges represent chemical bonds. Amazon Photos is a co-purchase network, and CS is a coauthor network linking researchers by joint publications.

**Victim Models.** Following prior work on link stealing attacks (He et al., 2021a; Zhang et al., 2023), we adopt the GCN (Kipf & Welling, 2016) as our primary victim model. The GCN is a two-layer architecture with a 16-dimensional hidden layer, ReLU activation, and a dropout rate of 0.5. To assess robustness across architectures, we also evaluate on GAT (Veličković et al., 2017) and GraphSAGE (Hamilton

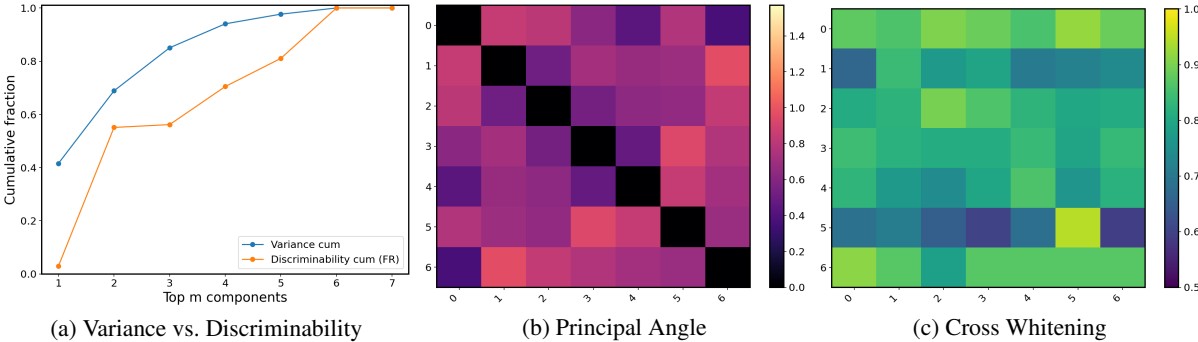

(a) Variance vs. Discriminability       (b) Principal Angle       (c) Cross Whitening

*Figure 3.* Intra-class discriminability analysis for the GAT model on Cora. (a) Variance and discriminability comparison. (b) Heatmap of principal angles between per-class discriminability subspaces. (c) AUC performance of the proposed cross-whitening strategy.

et al., 2017). The GAT is a two-layer model with eight attention heads, ELU activation, and a dropout rate of 0.5. The GraphSAGE model is a two-layer mean aggregator variant with a 16-dimensional hidden layer, ReLU activation, and a dropout rate of 0.5. All models are trained with cross-entropy loss, $L_2$ weight decay ($5 \times 10^{-4}$), the Adam optimizer (learning rate 0.01), and early stopping with patience 10.

**Evaluation Metrics.** We evaluate model effectiveness using the AUC, which quantifies average-case performance. Given a scoring model, the AUC corresponds to the probability that a randomly chosen positive instance is assigned a higher score than a randomly chosen negative instance. In addition to the average-case behavior, we also report the TPR at a fixed FPR of $10^{-3}$, as read from the ROC curve. This metric highlights performance in the low-FPR regime, which is particularly critical for intra-class pairs at the distributional tails, where the tolerance for errors is minimal.

**Distance Metrics.** We consider eight distance functions to compute the similarity between posteriors of node pairs: cosine, Euclidean, correlation, Chebyshev, Bray–Curtis, Canberra, cityblock, and squared Euclidean (He et al., 2021a; Zhang et al., 2023). Given the posterior outputs of the victim GNN, the attacker first collects the posterior vectors corresponding to each node pair of interest. The pairs are then partitioned into intra-class and inter-class groups based on predicted class labels, allowing the attack to analyze structural and semantic similarities separately. To mitigate class-level compression that can obscure fine-grained differences, the attacker applies PCW to the intra-class pairs before computing distances. Finally, the chosen distance measure is applied to compute pairwise similarity between posterior vectors. Smaller distances imply higher likelihoods of linkage, whereas larger distances indicate potential absence of connection.

### 6.2. Performance on Corrected Intra-Class Pairs

We evaluate how PCW improves link-stealing attacks on intra-class pairs, with emphasis on both global ranking

quality and reliability under strict low false-positive constraints. In Appendix E, we discuss the broader implications of our observations for evaluating and deploying link-stealing attacks in security-sensitive settings. Appendix F further provides a diagnostic analysis of posterior geometry and negative-score tail concentration, which explains when improvements in global ranking quality translate into actionable gains at low FPR, and when dataset-specific score geometry constrains such gains. Appendix H presents an analysis of the computational complexity of PCW relative to baseline distance-based attacks.

**Average Performance.** Table 2 summarizes AUC results across datasets and distance metrics on the GCN model. Since generic and uneven attacks adopt identical decision thresholds, their baseline results coincide; hence, we report the same values for both. In most cases, PCW yields substantial improvements over the generic attack. For instance, on the CS dataset, the AUC using correlation distance increases from 0.712 to 0.832. Canberra distance, which was previously weak on Cora (0.554), improves to 0.755 after whitening. The benefit extends beyond GCN: Tables 4 and 5 in the appendix show consistent trends for GAT and Graph-SAGE, demonstrating that PCW enhances attacks across diverse architectures.

**Performance at Low FPRs.** Average-case metrics alone can obscure practical risk, where only a small number of high-confidence successes matter. We therefore report TPR at a FPR of $10^{-3}$. PCW markedly strengthens the attack in this strict regime. For example, on the AIDS dataset, Braycurtis TPR increases from 3.5% to 6.5%, while Canberra improves from 1.5% to 4.2%. On citation graphs, Canberra TPR jumps from 10.9% to 19.3% on Cora, and from 18.1% to 23.0% on Citeseer. These gains confirm that our approach not only raises average separability but also exposes previously hidden vulnerabilities under realistic operating budgets.

*Table 2.* Performance of intra-class link–stealing attack using GCN. PCW indicates the per-class whitening.

| | | cosine | | euclidean | | sqeuclidean | | correlation | | cityblock | | chebyshev | | braycurtis | | canberra | |
|---|---|---|---|---|---|---|---|---|---|---|---|---|---|---|---|---|---|
| | | AUC | TPR | AUC | TPR | AUC | TPR | AUC | TPR | AUC | TPR | AUC | TPR | AUC | TPR | AUC | TPR |
| Cora | Intra | 0.699 | 0.203 | 0.631 | 0.184 | 0.631 | 0.184 | 0.747 | 0.229 | 0.640 | 0.146 | 0.603 | 0.143 | 0.640 | 0.146 | 0.554 | 0.109 |
| | Intra (PCW) | **0.790** | **0.227** | **0.717** | **0.201** | **0.717** | **0.201** | **0.862** | **0.238** | **0.705** | **0.244** | **0.724** | **0.163** | **0.786** | **0.218** | **0.755** | **0.193** |
| Citeseer | Intra | 0.762 | **0.200** | 0.694 | 0.176 | 0.694 | 0.176 | 0.835 | **0.221** | 0.699 | 0.200 | 0.669 | 0.143 | 0.699 | 0.200 | 0.647 | 0.181 |
| | Intra (PCW) | **0.830** | 0.154 | **0.807** | **0.335** | **0.807** | **0.335** | **0.895** | 0.169 | **0.791** | **0.366** | **0.814** | **0.316** | **0.812** | **0.328** | **0.742** | **0.230** |
| PubMed | Intra | 0.687 | 0.012 | 0.626 | **0.011** | 0.626 | **0.011** | **0.734** | 0.001 | **0.620** | **0.012** | 0.620 | **0.012** | 0.620 | 0.012 | 0.592 | 0.000 |
| | Intra (PCW) | **0.713** | **0.014** | **0.637** | 0.002 | **0.637** | 0.002 | 0.718 | **0.001** | 0.617 | 0.003 | **0.662** | 0.002 | **0.714** | **0.021** | **0.680** | **0.017** |
| AIDS | Intra | 0.689 | 0.031 | 0.662 | 0.034 | 0.662 | 0.034 | 0.691 | 0.031 | 0.677 | 0.035 | 0.643 | 0.032 | 0.677 | 0.035 | 0.596 | 0.015 |
| | Intra (PCW) | **0.770** | **0.066** | **0.752** | **0.098** | **0.752** | **0.098** | **0.779** | **0.069** | **0.733** | **0.086** | **0.760** | **0.048** | **0.731** | **0.065** | **0.674** | **0.042** |
| Photos | Intra | 0.674 | 0.013 | 0.576 | 0.011 | 0.576 | 0.011 | **0.850** | **0.027** | 0.589 | 0.012 | 0.540 | 0.009 | 0.589 | 0.012 | 0.520 | 0.012 |
| | Intra (PCW) | **0.760** | **0.024** | **0.700** | **0.019** | **0.700** | **0.019** | 0.795 | 0.025 | **0.690** | **0.018** | **0.710** | **0.027** | **0.757** | **0.027** | **0.715** | **0.025** |
| CS | Intra | 0.708 | 0.007 | 0.666 | 0.008 | 0.666 | 0.008 | 0.712 | 0.008 | **0.679** | 0.007 | 0.653 | 0.010 | 0.679 | 0.007 | 0.565 | **0.036** |
| | Intra (PCW) | **0.775** | **0.028** | **0.673** | **0.018** | **0.673** | **0.018** | **0.832** | **0.064** | 0.668 | **0.021** | **0.670** | **0.037** | **0.751** | **0.024** | **0.710** | 0.016 |

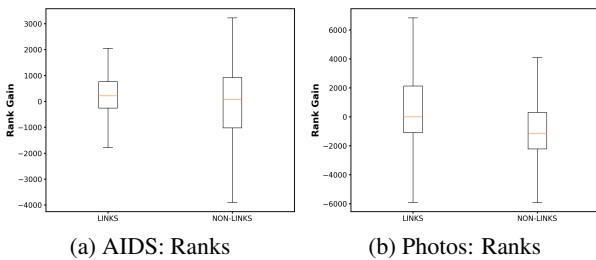

(a) AIDS: Ranks (b) Photos: Ranks

*Figure 4.* Rank-gain distribution of link and non-link pairs computed from distance scores for the AIDS and Photos datasets.

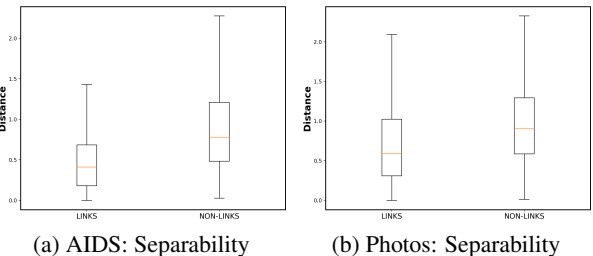

(a) AIDS: Separability (b) Photos: Separability

*Figure 5.* Boxplots of intra-pair distance scores for links and non-links on the AIDS and Photos datasets. The mean scores move away from zero, with links closer to 0.5 and non-links closer to 1, demonstrating clearer separation.

### 6.3. Ranking Gains for Links and Non-Links

We evaluate a link-stealing attack on intra-class node pairs to investigate whether an adversary can exploit the relative difficulty of these pairs to reorder node pairs. Fig. 4 reports the change in ranking for links and non-links based on Euclidean distance, where ranks are derived from the distance scores between node posteriors. Our hypothesis is that, for an approach to achieve good separability between links and non-links, it should reorder the candidate pairs so that non-links are pushed lower in the ranking relative to links. As seen in the results, non-links exhibit a larger loss in rank than links across both datasets, indicating that spurious connections become less competitive once distance-based ordering is applied. This effect is more pronounced in Photos, where the gap between link and non-link ranks is wider. These observations support the hypothesis that an attacker can exploit structural and posterior similarities between links and non-links to reorder candidate pairs.

### 6.4. Assessing the Separability of Links from Non-Links

To obtain a more detailed view of how well links can be distinguished from non-links, we analyzed the distribution of distance scores computed for intra-class node pairs. Figure 5 presents boxplots for the AIDS dataset (a) and the Photos dataset (b), using Euclidean distance as the similarity measure. Across both datasets, the mean distance for linked pairs shifts away from zero and concentrates around 0.5,

whereas non-linked pairs are distributed closer to 1. This displacement implies that, after applying the distance-based transformation, links occupy a region of the score space that is more clearly separated from spurious connections, indicating that the transformed distance scores yield a stronger contrast between the two groups. Such separation provides adversaries with a clearer decision boundary, enabling them to more accurately filter candidate pairs and thereby increase the likelihood of successful intra-class attacks.

## 7. Conclusion

In this work, we reexamine link-stealing attacks on GNNs through a geometric and tail-risk perspective. We identify anisotropic posterior collapse as the mechanism suppressing intra-class vulnerabilities by compressing discriminative structure and masking exposure. We then introduce a geometric correction that reconditions posteriors to amplify informative low-variance directions, restoring separability and exposing leakage even under strict precision constraints. More broadly, this work reframes link privacy as a geometric calibration problem and provides a diagnostic basis for geometry-aware, risk-sensitive privacy evaluation. Future work will extend this perspective to other structural partitions and defense settings, advancing a more principled understanding of graph-level privacy.

## Acknowledgements

We thank the anonymous reviewers for their comments and suggestions. This work has been partially supported by the National Science Foundation (NSF) under grant CNS-2443252. Any findings, conclusions, and recommendations expressed in this paper are those of the authors and do not necessarily reflect the views of the NSF.

## Impact Statement

While the proposed analysis and reconditioning techniques could be used to strengthen link inference attacks on GNNs, our defensive intent is to expose privacy failure modes that are overlooked by standard average-case metrics and to motivate more conservative evaluation practices and mitigations for deployed models. The broader societal impact of this work depends on deployment context, but we expect it to primarily benefit researchers and practitioners developing, evaluating, and governing machine-learning systems that operate on sensitive networked data.

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

*Table 3.* AUC and TPR@0.1% for link–nonlink discrimination on Citeseer and PubMed using eight distance metrics. Rows separate generic distances, inter-class and intra-class subsets, and CC-LS at two operating points $g_0$ (low margin) and $g_1$ (high margin). Bold entries denote the best score per block. Results highlight how confidence margin and class relation (inter vs. intra) interact with distance choice.

| | | cosine | | euclidean | | sqeuclidean | | correlation | | cityblock | | chebyshev | | braycurtis | | canberra | |
|---|---|---|---|---|---|---|---|---|---|---|---|---|---|---|---|---|---|
| | | AUC | TPR | AUC | TPR | AUC | TPR | AUC | TPR | AUC | TPR | AUC | TPR | AUC | TPR | AUC | TPR |
| Citeseer | Generic | 0.943 | 0.187 | 0.898 | 0.165 | 0.898 | 0.165 | 0.959 | 0.207 | 0.901 | 0.187 | 0.873 | 0.134 | 0.901 | 0.187 | 0.813 | 0.176 |
| | Inter | **0.934** | **0.294** | **0.902** | **0.294** | **0.902** | **0.294** | **0.951** | **0.324** | **0.907** | **0.412** | **0.882** | **0.265** | **0.907** | **0.412** | **0.872** | **0.353** |
| | Intra | 0.762 | 0.200 | 0.694 | 0.176 | 0.694 | 0.176 | 0.835 | 0.221 | 0.699 | 0.200 | 0.669 | 0.143 | 0.699 | 0.200 | 0.647 | 0.181 |
| | CC-LS ($g_0$) | 0.935 | 0.131 | 0.899 | 0.142 | 0.899 | 0.142 | 0.952 | 0.148 | 0.903 | 0.159 | 0.878 | 0.102 | 0.903 | 0.159 | **0.862** | **0.182** |
| | CC-LS ($g_1$) | **0.937** | 0.226 | **0.910** | 0.226 | **0.910** | 0.226 | **0.960** | **0.283** | **0.908** | **0.251** | **0.887** | **0.208** | **0.908** | **0.251** | 0.795 | 0.176 |
| | Inter—CC-LS ($g_0$) | 0.886 | 0.303 | 0.845 | **0.303** | 0.845 | **0.303** | 0.921 | 0.333 | 0.856 | 0.424 | 0.824 | 0.273 | 0.856 | 0.424 | 0.831 | 0.364 |
| | Inter—CC-LS ($g_1$) | **1.000** | **1.000** | **0.990** | 0.000 | **0.990** | 0.000 | **1.000** | **1.000** | **1.000** | **1.000** | **1.000** | **1.000** | **1.000** | **1.000** | **1.000** | **1.000** |
| | Intra—CC-LS ($g_0$) | **0.790** | 0.151 | **0.743** | 0.161 | **0.743** | 0.161 | **0.876** | 0.214 | **0.745** | 0.182 | **0.720** | 0.120 | **0.745** | 0.182 | **0.715** | **0.193** |
| | Intra—CC-LS ($g_1$) | 0.638 | **0.245** | 0.580 | **0.227** | 0.580 | **0.227** | 0.739 | **0.314** | 0.588 | **0.249** | 0.558 | **0.205** | 0.588 | **0.249** | 0.557 | 0.175 |
| Pubmed | Generic | 0.859 | 0.001 | 0.833 | 0.001 | 0.833 | 0.001 | 0.873 | 0.000 | 0.830 | 0.001 | 0.830 | 0.001 | 0.830 | 0.001 | 0.641 | 0.000 |
| | Inter | **0.830** | **0.008** | **0.801** | **0.008** | **0.801** | **0.008** | 0.816 | **0.011** | 0.805 | **0.013** | 0.805 | **0.013** | 0.805 | **0.013** | 0.742 | **0.011** |
| | Intra | 0.687 | 0.001 | 0.626 | 0.001 | 0.626 | 0.001 | 0.734 | 0.000 | 0.620 | 0.001 | 0.620 | 0.001 | 0.620 | 0.001 | 0.592 | 0.000 |
| | CC-LS ($g_0$) | 0.808 | 0.001 | 0.779 | 0.001 | 0.779 | 0.001 | 0.814 | 0.002 | 0.775 | 0.001 | 0.775 | 0.001 | 0.775 | 0.001 | 0.654 | 0.003 |
| | CC-LS ($g_1$) | 0.829 | 0.001 | 0.805 | 0.000 | 0.805 | 0.000 | 0.865 | 0.000 | 0.804 | 0.000 | 0.804 | 0.000 | 0.804 | 0.000 | 0.664 | 0.000 |
| | Inter—CC-LS ($g_0$) | **0.811** | **0.010** | **0.781** | **0.010** | **0.781** | **0.010** | 0.803 | 0.013 | **0.785** | **0.016** | **0.785** | **0.016** | **0.785** | **0.016** | **0.722** | 0.013 |
| | Inter—CC-LS ($g_1$) | 0.690 | 0.000 | 0.618 | 0.000 | 0.618 | 0.000 | 0.769 | 0.030 | 0.639 | 0.000 | 0.639 | 0.000 | 0.639 | 0.000 | 0.567 | **0.015** |
| | Intra—CC-LS ($g_0$) | **0.618** | 0.001 | **0.566** | 0.001 | **0.566** | 0.001 | **0.666** | 0.001 | **0.557** | 0.001 | **0.557** | 0.001 | **0.557** | 0.001 | 0.552 | 0.001 |
| | Intra—CC-LS ($g_1$) | 0.531 | **0.019** | 0.529 | **0.002** | **0.529** | 0.002 | 0.635 | **0.003** | 0.530 | **0.002** | 0.530 | **0.002** | 0.530 | **0.002** | **0.670** | **0.007** |

# A. Training and Predicting with GNN

**Training a GNN.** GNNs learn node embeddings by propagating and aggregating information over the graph structure. For a graph convolutional network (GCN) (Kipf & Welling, 2016), the update at layer $l$ is

$$\mathcal{Z}^{l+1} = \sigma\big(\tilde{\mathcal{D}}^{-1/2}\tilde{\mathcal{A}}\tilde{\mathcal{D}}^{-1/2}\mathcal{Z}^l\mathcal{W}^l\big), \tag{5}$$

where $\sigma(\cdot)$ is a nonlinear activation, $\tilde{\mathcal{A}} = \mathcal{A} + \mathcal{I}$ adds self-loops, $\tilde{\mathcal{D}}$ is its degree matrix, $\mathcal{W}^l$ is the weight matrix of the $l$-th layer, and $\mathcal{Z}^0 = \mathcal{X}$. The symmetric normalization $\tilde{\mathcal{D}}^{-1/2}\tilde{\mathcal{A}}\tilde{\mathcal{D}}^{-1/2}$ ensures balanced message passing, preventing nodes with large degrees from dominating aggregation and stabilizes training. Beyond GCNs, other architectures modify the aggregation mechanism. Graph Attention Networks (GATs) assign learnable attention coefficients to weight neighbors dynamically (Veličković et al., 2017). GraphSAGE (Hamilton et al., 2017) samples a fixed-size neighborhood and aggregates via mean, max, or LSTM functions. These alternatives increase expressiveness and adapt GNNs to diverse data regimes.

**Prediction with Learned Embeddings.** The final embedding $\mathcal{Z}^L$ produced by a GNN serves as input to downstream models. For node classification, a simple linear layer followed by a Softmax yields a probability vector over $C$ target classes:

$$\widetilde{Y} = \text{Softmax}(Z^L W_{\text{out}}). \tag{6}$$

where $\tilde{y} \in \mathbb{R}^{|\mathcal{V}| \times C}$, and the $i$-th row gives the predicted class probabilities for node $i$.

The posteriors produced by the GNN, $\tilde{\mathcal{y}}$, are often accessible in deployed systems via prediction APIs. In this work, we adopt a black-box threat model where an adversary can query the target GNN and obtain posterior probabilities $p(u)$ for individual nodes. The attacker's objective is to infer whether a link $(u, v)$ exists in the training graph based solely on these posteriors. We next formalize this link-stealing process.

# B. Evaluating Generic Link-Stealing Attacks: Citeseer and PubMed

In Table 3, we show the result of our benchmark for the Citeseer and the PubMed datasets.

# C. Cosine-based link prediction investigation for Citeseer and PubMed

In Fig. 6, we show the results for cosine-based link prediction investigation for Citeseer and Pubmed datasets.

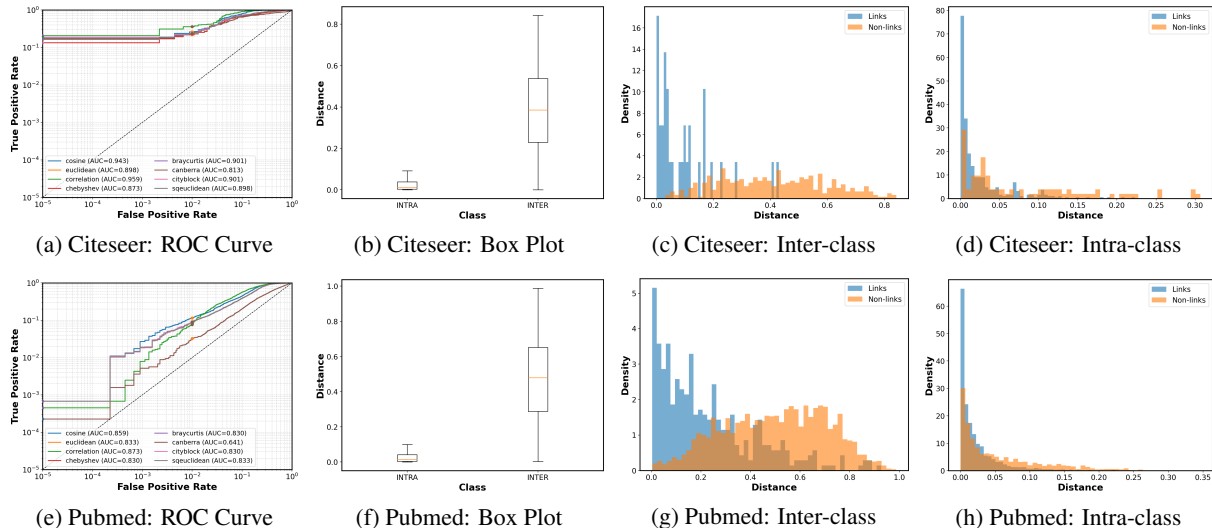

(a) Citeseer: ROC Curve  (b) Citeseer: Box Plot  (c) Citeseer: Inter-class  (d) Citeseer: Intra-class

(e) Pubmed: ROC Curve  (f) Pubmed: Box Plot  (g) Pubmed: Inter-class  (h) Pubmed: Intra-class

*Figure 6.* Cosine-based link prediction investigation across Citeseer and PubMed. We report ROC curves, box plots, and cosine similarity distributions for inter-class and intra-class node pairs.

*Table 4.* Performance of intra-class link–stealing attack using GAT. PCW indicates the per-class whitening.

|  |  | cosine | | euclidean | | sqeuclidean | | correlation | | cityblock | | chebyshev | | braycurtis | | canberra | |
|---|---|---|---|---|---|---|---|---|---|---|---|---|---|---|---|---|---|
|  |  | AUC | TPR | AUC | TPR | AUC | TPR | AUC | TPR | AUC | TPR | AUC | TPR | AUC | TPR | AUC | TPR |
| Cora | Intra | 0.791 | 0.104 | 0.818 | 0.087 | 0.818 | 0.087 | 0.761 | 0.179 | 0.827 | 0.112 | 0.799 | 0.084 | 0.827 | 0.112 | 0.859 | **0.315** |
|  | Intra (PCW) | **0.904** | **0.203** | **0.874** | **0.294** | **0.874** | **0.294** | **0.920** | **0.479** | **0.869** | **0.322** | **0.866** | **0.168** | **0.894** | **0.380** | **0.862** | 0.292 |
| Citeseer | Intra | 0.907 | 0.255 | 0.908 | 0.342 | 0.908 | 0.342 | 0.880 | **0.252** | 0.915 | 0.326 | 0.892 | 0.323 | 0.915 | 0.326 | **0.928** | **0.365** |
|  | Intra (PCW) | **0.937** | **0.466** | **0.940** | **0.424** | **0.940** | **0.424** | **0.941** | 0.211 | **0.930** | **0.424** | **0.939** | **0.424** | **0.926** | **0.417** | 0.883 | 0.303 |
| PubMed | Intra | 0.817 | 0.004 | 0.829 | **0.009** | 0.829 | **0.009** | 0.770 | 0.003 | 0.830 | **0.010** | 0.830 | **0.010** | 0.830 | 0.010 | **0.860** | 0.012 |
|  | Intra (PCW) | **0.834** | **0.007** | **0.858** | 0.007 | **0.858** | 0.007 | **0.834** | **0.010** | **0.857** | 0.007 | **0.858** | 0.006 | **0.847** | **0.021** | 0.829 | **0.017** |
| AIDS | Intra | 0.873 | 0.025 | 0.874 | 0.024 | 0.874 | 0.024 | 0.872 | 0.025 | 0.883 | 0.018 | 0.871 | 0.021 | 0.883 | 0.018 | 0.873 | 0.024 |
|  | Intra (PCW) | **0.920** | **0.073** | **0.914** | **0.052** | **0.914** | **0.052** | **0.913** | **0.103** | **0.912** | **0.060** | **0.901** | **0.081** | **0.921** | **0.087** | **0.904** | **0.067** |
| Photos | Intra | 0.898 | 0.050 | 0.896 | 0.051 | 0.896 | 0.051 | **0.893** | 0.037 | 0.903 | 0.052 | 0.887 | 0.053 | **0.903** | 0.052 | **0.912** | 0.049 |
|  | Intra (PCW) | **0.899** | **0.053** | **0.917** | 0.051 | **0.917** | 0.051 | 0.887 | **0.055** | **0.909** | 0.052 | **0.916** | **0.056** | 0.884 | **0.061** | 0.826 | **0.069** |
| CS | Intra | 0.774 | 0.022 | 0.823 | 0.045 | 0.823 | 0.045 | 0.739 | 0.021 | **0.831** | 0.044 | 0.809 | 0.050 | 0.831 | 0.044 | 0.849 | 0.100 |
|  | Intra (PCW) | **0.896** | **0.092** | **0.830** | **0.102** | **0.830** | **0.102** | **0.899** | **0.145** | 0.829 | **0.078** | **0.818** | **0.099** | **0.884** | **0.138** | 0.835 | **0.131** |

## D. Results of PCW Attack on GAT and GraphSAGE

This section presents additional results of the PCW attack on GAT and GraphSAGE models. As reported in Table 4 and Table 5, PCW consistently enhances intra-class link-stealing performance, mirroring the gains observed for GCN. For example, on the Cora dataset, using the correlation distance increases AUC from 0.761 to 0.920 and raises TPR from 0.179 to 0.479. Also, for GraphSAGE on the CS dataset, correlation-based attacks achieve an AUC improvement from 0.605 to 0.823, with TPR rising from 0.0006 to 0.023. These results confirm that the benefits of whitening extend across different GNN architectures, strengthening the attack's effectiveness in most settings.

## E. Implications

The empirical results of our attack reveal important consequences for both adversaries and defenders in graph-based systems. By exposing additional weaknesses in intra-class links and enhancing attack success under strict false-positive budgets, PCW reshapes the understanding of link privacy in GNNs.

**Link Vulnerability.** Our findings indicate that a substantially larger fraction of links is vulnerable to inference than previously reported. This expanded attack surface implies that adversaries, even with only black-box access, can recover more structural information than expected. By incorporating PCW, an attacker can bypass limitations that previously limited the success of intra-class pairs, enabling broader and more precise reconstruction of graph topology.

*Table 5.* Performance of intra-class link–stealing attack using GraphSAGE. PCW indicates the per-class whitening introduced to improve the intra-class attack performance.

| | | cosine | | euclidean | | sqeuclidean | | correlation | | cityblock | | chebyshev | | braycurtis | | canberra | |
| | | AUC | TPR | AUC | TPR | AUC | TPR | AUC | TPR | AUC | TPR | AUC | TPR | AUC | TPR | AUC | TPR |
|---|---|---|---|---|---|---|---|---|---|---|---|---|---|---|---|---|---|
| Cora | Intra | 0.692 | **0.073** | 0.724 | 0.095 | 0.724 | 0.095 | 0.679 | 0.070 | 0.730 | 0.090 | 0.715 | 0.126 | 0.730 | 0.090 | **0.805** | **0.176** |
| | Intra (PCW) | **0.827** | 0.062 | **0.781** | **0.163** | **0.781** | **0.163** | **0.825** | **0.233** | **0.786** | 0.165 | **0.767** | **0.137** | **0.833** | **0.097** | 0.794 | 0.048 |
| Citeseer | Intra | 0.823 | 0.090 | 0.852 | 0.100 | 0.852 | 0.100 | 0.792 | 0.122 | 0.857 | 0.112 | 0.849 | 0.061 | 0.857 | 0.112 | **0.887** | **0.364** |
| | Intra (PCW) | **0.907** | **0.227** | **0.901** | **0.249** | **0.901** | **0.249** | **0.907** | **0.308** | **0.898** | **0.213** | **0.889** | **0.215** | **0.905** | **0.232** | 0.886 | 0.264 |
| PubMed | Intra | 0.752 | **0.024** | 0.766 | 0.019 | 0.766 | 0.019 | 0.717 | **0.004** | 0.769 | **0.021** | 0.769 | **0.021** | 0.769 | **0.021** | **0.830** | **0.034** |
| | Intra (PCW) | **0.799** | 0.011 | **0.804** | **0.021** | **0.804** | **0.021** | **0.798** | 0.003 | **0.806** | 0.020 | **0.801** | 0.016 | **0.808** | 0.023 | 0.790 | 0.030 |
| AIDS | Intra | 0.701 | 0.015 | 0.705 | 0.011 | 0.705 | 0.011 | 0.701 | 0.015 | 0.709 | 0.015 | 0.706 | 0.011 | 0.709 | 0.015 | **0.804** | 0.031 |
| | Intra (PCW) | **0.751** | **0.070** | **0.733** | **0.061** | **0.733** | **0.061** | **0.742** | **0.070** | **0.746** | **0.064** | **0.720** | **0.100** | **0.775** | **0.077** | 0.793 | **0.070** |
| Photos | Intra | **0.841** | 0.013 | 0.820 | **0.010** | 0.820 | **0.010** | **0.835** | 0.024 | 0.828 | 0.012 | 0.798 | 0.008 | **0.828** | 0.012 | **0.802** | 0.009 |
| | Intra (PCW) | 0.813 | **0.025** | **0.831** | 0.009 | **0.831** | 0.009 | 0.808 | **0.031** | **0.830** | 0.009 | **0.830** | 0.009 | 0.803 | **0.019** | 0.768 | **0.027** |
| CS | Intra | 0.614 | 0.006 | 0.666 | 0.005 | 0.666 | 0.005 | 0.605 | 0.006 | 0.664 | 0.008 | 0.664 | 0.004 | 0.663 | 0.008 | **0.760** | 0.023 |
| | Intra (PCW) | **0.813** | **0.010** | **0.726** | **0.020** | **0.726** | **0.020** | **0.823** | **0.023** | **0.746** | **0.017** | **0.719** | **0.012** | **0.793** | **0.017** | 0.741 | **0.005** |

**Success at Low FPRs.** The gains at low FPR reveal that certain links remain highly exposed even when an adversary adopts a conservative budget for errors. From a defender's standpoint, these worst-case instances represent critical weak points in the graph: a small subset of edges that can be inferred with high confidence despite stringent detection thresholds. Identifying and mitigating such high-risk links through access control, noise injection, or structural regularization should therefore be a priority for protecting graph data against link-stealing attacks.

# F. Diagnostic Analysis of Posterior Geometry and Tail Reliability

Across datasets, our method consistently improves global ranking quality, as reflected by gains in AUC. However, on PubMed, these improvements do not uniformly translate to higher TPR at extreme low FPRs, even when AUC increases. Since such operating points are critical for security-relevant deployment, we analyze why the underlying score geometry of certain datasets may limit gains in this regime. Our analysis focuses on identifying dataset-dependent geometric factors that govern behavior at very low FPR.

**Posterior smoothness.** For each node, the GNN outputs a class posterior $p \in \mathbb{R}^K$. We quantify posterior concentration using the effective posterior support, defined as

$$\exp(H(p)), \quad \text{where} \quad H(p) = -\sum_{k=1}^{K} p_k \log p_k$$

is the Shannon entropy. This quantity admits a direct interpretation as the effective number of classes receiving non-negligible probability mass. Lower values indicate sharper and more confident predictions, whereas higher values correspond to more diffuse posteriors.

Since link-stealing scores are defined on node pairs, we compute this statistic on the pair-averaged posterior $\frac{1}{2}(p_0 + p_1)$ and report the mean across all test pairs for each dataset.

**Negative tail spread.** To analyze reliability at extreme low-FPR operating points, we examine the distribution of directed attack scores for negative *intra* pairs, where higher scores indicate higher confidence in a link. Since low-FPR thresholds are governed by the extreme upper tail of the negative score distribution, we measure tail concentration as

$$q_{0.999} - q_{0.99},$$

the spread between the 99.9th and 99th percentiles of negative scores. Smaller values indicate a more concentrated extreme tail, in which a small number of negatives tightly define the decision threshold.

**Results.** Figure 7(a) reports the mean effective posterior support across datasets. PubMed exhibits substantially lower effective support than Cora and CiteSeer, indicating that its posteriors are significantly more peaked and confident. This reduced posterior variability induces a score geometry in which only a small number of negative *intra* pairs lie near the decision boundary. Figure 7(b) further quantifies this effect by reporting the negative tail concentration (NTS), measured

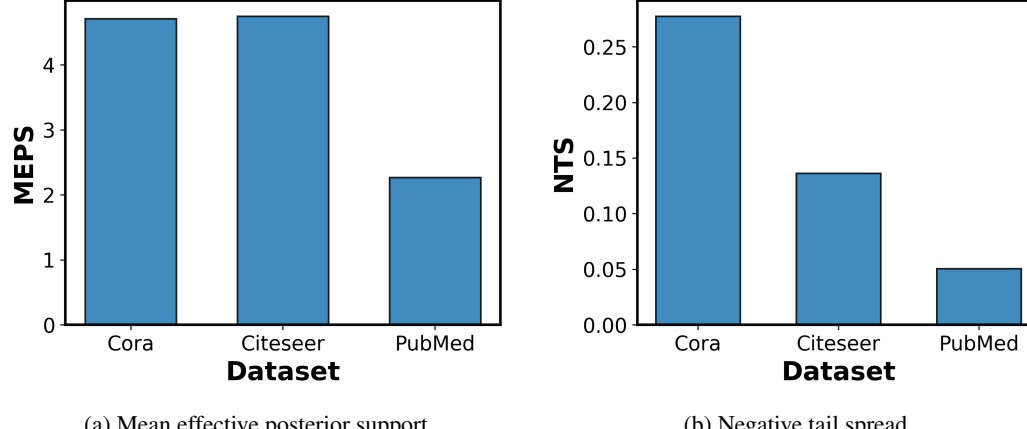

(a) Mean effective posterior support

(b) Negative tail spread

*Figure 7.* Posterior and tail diagnostics across datasets. (a) MEPS, showing that PubMed exhibits more peaked and confident predictions. (b) NTS, showing that PubMed has a substantially more concentrated extreme tail. These geometric differences explain why gains in global ranking metrics may not translate to improved TPR at extreme low-FPR operating points.

*Table 6.* Large-scale evaluation on OGBN-Products. Entries report AUC / TPR@0.1% / TPR@1%.

| Model | Setting | Cos | Eucl | SqEuc | Corr | City | Cheb | Bray | Canb |
|---|---|---|---|---|---|---|---|---|---|
| GCN | Intra | 0.573/0.027/0.059 | 0.517/0.000/0.003 | 0.517/0.000/0.003 | 0.593/0.106/0.179 | 0.542/0.019/0.041 | 0.562/0.000/0.003 | 0.542/0.019/0.041 | 0.573/0.000/0.009 |
|  | PCW | **0.777/0.161/0.302** | **0.682/0.081/0.175** | **0.682/0.081/0.175** | **0.784/0.160/0.301** | **0.667/0.068/0.150** | **0.682/0.083/0.180** | **0.756/0.117/0.252** | **0.713/0.073/0.159** |
| GAT | Intra | 0.613/0.002/0.015 | 0.631/0.002/0.015 | 0.631/0.002/0.015 | 0.611/0.002/0.015 | 0.637/0.002/0.015 | 0.628/0.002/0.015 | 0.637/0.002/0.015 | 0.645/**0.022/0.072** |
|  | PCW | **0.766**/0.003/0.028 | **0.695**/0.002/0.021 | **0.695**/0.002/0.021 | **0.771**/0.003/0.031 | **0.692**/0.002/0.021 | **0.684**/0.002/0.021 | **0.750**/0.003/0.026 | **0.731**/0.004/0.041 |
| SAGE | Intra | **0.650**/0.006/0.047 | 0.619/0.006/0.059 | 0.619/0.006/0.059 | **0.645**/0.006/0.047 | 0.638/0.006/0.056 | **0.610**/0.006/0.054 | 0.638/0.006/0.056 | 0.547/**0.018**/0.044 |
|  | PCW | 0.646/**0.007/0.089** | **0.634**/0.007/0.086 | **0.634**/0.007/0.086 | 0.645/**0.007/0.092** | **0.644**/0.007/0.079 | 0.606/**0.007/0.080** | **0.640**/0.007/0.087 | **0.619**/0.006/**0.077** |

as $q_{0.999} - q_{0.99}$. PubMed exhibits a markedly more concentrated extreme tail than the other datasets, implying that the very-low-FPR operating point is determined by a small and tightly clustered set of borderline negative pairs.

These diagnostics explain why improvements in global ranking metrics can coexist with reduced TPR at extreme low FPR on PubMed. When the negative tail is highly concentrated, small redistributions in score geometry, although beneficial on average, can reorder the few negatives that define the low-FPR threshold. As a result, tail recall may decrease even as overall separability improves. In contrast, datasets such as Cora and CiteSeer exhibit broader negative tails, where similar redistributions strengthen global ranking without materially affecting tail reliability. Importantly, this behavior does not undermine the relevance of low-FPR evaluation. On the contrary, it reinforces its necessity. Global metrics alone would suggest uniformly stronger attacks, whereas low-FPR analysis reveals dataset-dependent limits on actionable exploitability that are critical in security-sensitive settings. The PubMed case demonstrates that attack strength must be assessed not only by average separability, but by whether improvements persist at operationally meaningful false-positive rates.

## G. Large-Scale Evaluation of PCW

We evaluate PCW on the large-scale OGBN-Products dataset (Hu et al., 2020) to assess whether the method remains effective under realistic graph sizes and highly imbalanced pair distributions. Table 6 reports AUC, TPR@0.1%, and TPR@1% across multiple distance metrics and GNN architectures. Overall, PCW consistently improves performance over the intra-class baseline. For example, under cosine distance, GCN improves from 0.573/0.027/0.059 to 0.777/0.161/0.302. Improvements are especially pronounced at low-FPR operating points, where the baseline often struggles to separate positive and negative pairs. These results demonstrate that PCW scales beyond small citation graphs and remains effective on realistic graphs containing millions of nodes and pair comparisons.

## H. Computational Complexity.

We analyze the computational cost of the proposed PCW and scoring procedure and compare it to the baseline distance-based attack. Let $N$ denote the number of evaluated node pairs and $K$ the posterior dimension. For the baseline attack, scoring a single pair consists of computing a distance metric between two $K$-dimensional posterior vectors, which requires $O(K)$ time.

*Table 7.* Timing comparison of baseline Euclidean attack and PCW (per-class whitening) across datasets. Fit time denotes the one-time preprocessing cost for PCW, while attack time corresponds to per-pair scoring. Total time for PCW is the sum of fit and attack times.

| Dataset | #Pairs | Baseline (s) | PCW Fit (s) | PCW Attack (s) | PCW Total (s) |
|---|---|---|---|---|---|
| Cora | 1,056 | 0.032 | 0.034 | 0.035 | 0.069 |
| Citeseer | 912 | 0.028 | 0.030 | 0.031 | 0.061 |
| PubMed | 8,866 | 0.265 | 0.268 | 0.410 | 0.678 |
| AIDS | 6,478 | 0.199 | 0.213 | 0.238 | 0.451 |
| Photos | 23,818 | 0.745 | 0.742 | 0.829 | 1.571 |
| CS | 16,380 | 0.492 | 0.537 | 0.547 | 1.085 |
| OGBN-Products | 12,371,804 | 399.571 | 465.576 | 495.305 | 960.881 |

*Table 8.* Performance of intra-class link-stealing attack on Cora under different pair fractions.

| | | cosine | | euclidean | | sqeuclidean | | correlation | | cityblock | | chebyshev | | braycurtis | | canberra | |
|---|---|---|---|---|---|---|---|---|---|---|---|---|---|---|---|---|---|---|
| | | AUC | TPR | AUC | TPR | AUC | TPR | AUC | TPR | AUC | TPR | AUC | TPR | AUC | TPR | AUC | TPR |
| Full | Intra | 0.791 | 0.104 | 0.818 | 0.087 | 0.818 | 0.087 | 0.761 | 0.179 | 0.827 | 0.112 | 0.799 | 0.084 | 0.827 | 0.112 | 0.859 | **0.315** |
| | Intra (PCW) | **0.904** | **0.203** | **0.874** | **0.294** | **0.874** | **0.294** | **0.920** | **0.479** | **0.869** | **0.322** | **0.866** | **0.168** | **0.894** | **0.380** | **0.862** | 0.292 |
| 0.1 | Intra | 0.748 | 0.133 | 0.767 | 0.111 | 0.767 | 0.111 | 0.730 | 0.289 | 0.789 | 0.111 | 0.731 | 0.044 | 0.789 | 0.111 | **0.833** | 0.289 |
| | Intra (PCW) | **0.913** | **0.489** | **0.881** | **0.400** | **0.881** | **0.400** | **0.854** | 0.289 | **0.854** | **0.311** | **0.891** | **0.578** | **0.867** | **0.311** | 0.785 | **0.378** |
| 0.3 | Intra | 0.838 | 0.353 | 0.847 | 0.338 | 0.847 | 0.338 | 0.825 | 0.272 | 0.854 | 0.294 | 0.824 | 0.257 | 0.854 | 0.294 | 0.864 | **0.412** |
| | Intra (PCW) | **0.926** | **0.500** | **0.872** | **0.500** | **0.872** | **0.500** | **0.932** | **0.382** | **0.867** | **0.515** | **0.871** | **0.529** | **0.925** | **0.522** | **0.898** | 0.382 |
| 0.5 | Intra | 0.809 | 0.191 | 0.837 | 0.212 | 0.837 | 0.212 | 0.771 | 0.280 | 0.841 | 0.195 | 0.824 | 0.208 | 0.841 | 0.195 | **0.853** | **0.398** |
| | Intra (PCW) | **0.903** | **0.479** | **0.885** | **0.483** | **0.885** | **0.483** | **0.909** | **0.470** | **0.872** | **0.352** | **0.887** | **0.606** | **0.882** | **0.411** | 0.837 | 0.326 |

Evaluating $N$ pairs therefore has time complexity $O(NK)$. For the PCW, each endpoint posterior is first transformed using a dense $K \times K$ whitening matrix before the distance is computed. This introduces a per-pair cost of $O(K^2)$, dominated by the matrix–vector multiplications; the subsequent distance computation is $O(K)$ and does not change the asymptotic order. As a result, scoring $N$ intra-class pairs has time complexity $O(NK^2)$. Notably, the baseline and PCW scale linearly with the number of evaluated pairs $N$. The difference in computational cost arises from the dependence on the posterior dimension $K$. In typical node classification benchmarks, $K$ corresponds to the number of classes and is relatively small, so the additional $K$-factor introduced by whitening constitutes a modest constant-factor overhead in practice.

**Empirical Timing Analysis.** Table 7 reports the runtime of the baseline Euclidean attack and the proposed PCW method across datasets. We report the total PCW runtime, defined as the sum of a one-time fitting cost and a per-pair attack cost. From an attacker's perspective, the proposed method remains computationally feasible across all scales. Although PCW introduces additional overhead compared to the baseline, this cost is structured and practical. The fitting stage is a one-time preprocessing step used to compute per-class transformations, and can be reused across multiple attack evaluations. The attack stage, which operates on individual pairs, scales linearly with the dataset size. Even at large scale, the attack remains efficient. On OGBN-Products (12.37M pairs), PCW completes in approximately 961 seconds, corresponding to a per-pair cost of $\sim 40, \mu s$ and demonstrating that an adversary can execute the attack over millions of samples within minutes.

## I. Performance under Limited Samples and Truncated Posteriors

**Is the attack effective with limited pairs?** A potential concern is that PCW may require large numbers of pair samples to estimate class-conditional covariance reliably, particularly for rare classes. However, posterior vectors are concentrated in a low-dimensional effective space rather than occupying the full simplex uniformly. As a result, the covariance structure can still be estimated with substantially reduced sample counts.

Table 8 evaluates attack performance under different pair fractions on Cora. Across all sampling regimes, PCW consistently outperforms the corresponding intra-class baseline across most distance metrics. Importantly, the gains remain stable even when only 10% of pair samples are retained. For example, under cosine distance, PCW improves TPR from 0.133 to 0.489 at 10% sampling, from 0.353 to 0.500 at 30% sampling, and from 0.191 to 0.479 at 50% sampling.

**Is the attack effective with limited posterior information?** We next evaluate whether PCW remains effective when the attacker only observes truncated posterior vectors. Table 9 reports TPR under Top-2 and Top-3 posterior truncation settings.

*Table 9.* Cora comparison between Intra and Intra (PCW) under Top-2 and Top-3 settings.

| | Top-2 | | | Top-3 | | |
|---|---|---|---|---|---|---|
| **Metric** | **Intra** | **PCW** | **Gain** | **Intra** | **PCW** | **Gain** |
| cosine | 0.043 | **0.045** | +0.002 | 0.039 | **0.054** | +0.015 |
| euclidean | 0.043 | **0.052** | +0.009 | 0.039 | **0.043** | +0.004 |
| correlation | 0.043 | **0.045** | +0.002 | 0.039 | **0.054** | +0.015 |
| chebyshev | 0.043 | **0.052** | +0.009 | 0.041 | **0.043** | +0.002 |
| braycurtis | 0.043 | **0.050** | +0.007 | 0.041 | **0.050** | +0.009 |
| canberra | **0.052** | 0.043 | -0.009 | **0.058** | 0.048 | -0.010 |
| cityblock | 0.043 | **0.052** | +0.009 | 0.041 | **0.048** | +0.007 |
| sqeuclidean | 0.043 | **0.052** | +0.009 | 0.039 | **0.043** | +0.004 |

*Table 10.* Ablation study of distance metrics and normalization strategies.

| **Method** | **Raw / $p^\tau$** | **Centering** | **Covariance** | **Class-cond.** |
|---|---|---|---|---|
| Raw posterior distance | raw | no | none | no |
| Power only | $p^\tau$ | no | none | no |
| Class-centered only | $p^\tau$ | yes | none | yes |
| Diagonal whitening | $p^\tau$ | yes | diag | yes |
| Global full whitening | $p^\tau$ | yes | full | no |
| PCW | $p^\tau$ | yes | full | yes |

Under Top-2 and Top-3 truncation, PCW consistently improves over the intra-class baseline for most distance metrics. For example, under cosine distance, PCW improves TPR from 0.043 to 0.045 in the Top-2 setting and from 0.039 to 0.054 in the Top-3 setting. However, we observed that when only the top-1 posterior entry is retained, both methods collapse to near-random performance (TPR $\approx 0.001$). This result indicates that PCW fundamentally depends on structured posterior geometry rather than scalar confidence magnitude alone. While moderate truncation still preserves enough covariance structure for whitening to remain effective, extreme truncation destroys most relational information between posterior coordinates, causing both the baseline and PCW to fail.

## J. Impact of Covariance Reconditioning

We perform an ablation using different methods as shown in Table 10 to distinguish whether the improvements in our results arise from simple posterior rescaling or from the reconditioning performed by PCW. Table 11 reports the corresponding Intra-group results on Cora using GAT across multiple distance metrics. The results show that PCW's class-conditional reconditioning provides the largest additional improvement over using raw posterior values.

We next analyze the sensitivity of PCW to key design choices and hyperparameters, including the power exponent, covariance mode, query budget, temperature scaling, and class imbalance ratio. Table 12 evaluates the sensitivity of PCW to key design choices and hyperparameters. Varying the power exponent produces relatively stable performance (AUC 0.841–0.872), indicating that PCW is not highly sensitive to the exact transformation strength. Similarly, covariance modeling substantially affects performance, with Ledoit–Wolf shrinkage (Ledoit & Wolf, 2004) outperforming diagonal covariance estimation (0.855 vs. 0.812 AUC), highlighting the importance of accurately capturing posterior covariance structure. Query budget has comparatively limited impact once moderate numbers of queries are available, suggesting that PCW remains effective under constrained query sizes. Temperature scaling provides small but consistent improvements, while varying the class imbalance ratio causes only minor fluctuations. The results indicate that PCW is robust across a wide range of hyperparameter settings.

## K. Robustness and Defense Implications

We evaluate PCW under temperature scaling, Gaussian noise, Laplace noise, and differential privacy (Dwork, 2025). Table 13 reports representative results on Citeseer using GAT and cosine distance. Performance degrades as perturbation strength increases, especially under stronger Gaussian, Laplace, and DP noise. However, PCW consistently outperforms the baseline at low-FPR operating points. For example, under Gaussian noise with variance 0.1, the baseline achieves TPR of 0.042, whereas PCW retains 0.235. Under Laplace noise with scale 0.1, PCW improves TPR from 0.050 to 0.273.

*Table 11.* Intra-group results for Cora across distance metrics and ablation variants. For each method, the highest AUC and highest TPR@0.1% across distance metrics are shown in bold.

| Distance | Raw posterior | | Power only | | Class-centered | | Diagonal whitening | | Global full whitening | | PCW | |
|---|---|---|---|---|---|---|---|---|---|---|---|---|
| | AUC | TPR | AUC | TPR | AUC | TPR | AUC | TPR | AUC | TPR | AUC | TPR |
| cosine | 0.791 | 0.104 | 0.858 | 0.244 | 0.809 | 0.099 | 0.818 | 0.095 | 0.856 | 0.160 | **0.904** | **0.203** |
| euclidean | 0.818 | 0.0864 | 0.858 | 0.244 | 0.858 | 0.244 | 0.849 | 0.222 | 0.841 | 0.147 | **0.874** | **0.294** |
| sqeuclidean | 0.818 | 0.086 | 0.858 | 0.244 | 0.858 | 0.244 | 0.849 | 0.222 | 0.841 | 0.147 | **0.874** | **0.294** |
| correlation | 0.761 | 0.179 | 0.807 | 0.242 | 0.813 | 0.104 | 0.837 | 0.110 | 0.853 | 0.311 | **0.920** | **0.479** |
| cityblock | 0.827 | 0.112 | 0.851 | 0.214 | 0.851 | 0.214 | 0.838 | 0.240 | 0.825 | 0.149 | **0.869** | **0.323** |
| chebyshev | 0.799 | 0.084 | 0.863 | 0.309 | 0.863 | **0.309** | 0.856 | 0.229 | 0.852 | 0.149 | **0.866** | 0.168 |
| braycurtis | 0.827 | 0.112 | 0.856 | 0.225 | 0.802 | 0.138 | 0.805 | 0.151 | 0.841 | 0.108 | **0.894** | **0.380** |
| canberra | 0.859 | **0.315** | 0.860 | 0.315 | 0.773 | 0.130 | 0.773 | 0.130 | 0.799 | 0.093 | **0.862** | 0.292 |

*Table 12.* Sensitivity analysis of PCW components on Citeseer using GAT and cosine distance.

| | Power exponent ($\tau$) | | | | Query budget ($Q$) | | | | Covariance mode ($C$) | | | | Temperature ($T$) | | | |
|---|---|---|---|---|---|---|---|---|---|---|---|---|---|---|---|---|
| Metric | 0.25 | 0.5 | 0.75 | 1.0 | 32 | 64 | 128 | 256 | Diag. | None | Ledoit | Floor | 1 | 1.5 | 2 | 3 |
| AUC | 0.857 | 0.849 | 0.841 | **0.872** | 0.830 | 0.843 | **0.848** | 0.848 | 0.812 | 0.849 | **0.855** | 0.849 | 0.849 | 0.860 | 0.865 | **0.871** |
| TPR | **0.328** | 0.320 | 0.312 | 0.300 | 0.285 | 0.292 | **0.320** | 0.314 | 0.270 | 0.320 | **0.328** | 0.321 | 0.320 | 0.326 | 0.330 | **0.333** |

| | Imbalance ratio ($I$) | | | |
|---|---|---|---|---|
| Metric | 0.25 | 0.5 | 0.75 | 1.0 |
| AUC | 0.839 | 0.845 | 0.846 | **0.849** |
| TPR | 0.318 | **0.325** | 0.314 | 0.320 |

Moderate temperature scaling also preserves strong attack performance, with $T = 1.5$ achieving 0.942/0.518. While stronger perturbations can degrade the discriminative information required for downstream utility, our results show that mild perturbations fail to fully eliminate the leakage signal.

*Table 13.* Defense evaluation on Citeseer using GAT and cosine distance.

| Defense | Value | Base AUC | Base TPR | PCW AUC | PCW TPR |
|---|---|---|---|---|---|
| None | 0 | 0.907 | 0.255 | **0.937** | **0.466** |
| Temperature | 0.7 | 0.885 | 0.213 | **0.933** | **0.353** |
| Temperature | 1.5 | 0.921 | 0.317 | **0.942** | **0.518** |
| Gaussian | 0.05 | 0.901 | 0.120 | **0.909** | **0.159** |
| Gaussian | 0.1 | **0.871** | 0.042 | 0.849 | **0.235** |
| Laplace | 0.05 | 0.892 | 0.119 | **0.901** | **0.241** |
| Laplace | 0.1 | **0.859** | 0.050 | 0.844 | **0.273** |
| Differential Privacy | 10 | **0.859** | 0.050 | 0.844 | **0.273** |
| Differential Privacy | 5 | **0.773** | 0.020 | 0.729 | **0.038** |

