# OpenReview forum: "Revisiting Asymmetries in Black-box Link Stealing against Graph Neural Networks"
_ICML.cc/2026/Conference — ICML 2026 regular_

### Official Review · Reviewer_eWSS · 2026-03-12

**Soundness:** 3
**Presentation:** 2
**Significance:** 3
**Originality:** 2
**Overall Recommendation:** 4
**Confidence:** 2

**Summary:**

The paper works on Membership Inference Attacks (MIA) for Graph Neural Networks (GNNs). Specifically, they revisit and provide new insights for link-stealing attacks. In particular, they focus on settings with extremely low false positive rates, revealing tail-risk vulnerabilities. They propose a new method, Per-Class Whitening (PCW), that reconditions the geometry of posteriors to recover the structure.

**Compliance With Llm Reviewing Policy:**

Affirmed.

**Final Justification:**

The rebuttal answers addressed my main concerns. I maintain my positive score.

**Key Questions For Authors:**

- Can you include a statistical analysis on how significant the proposed method's improvement is compared to the baseline?
- Can you propose a direction of research to develop a defense against this newly discovered attack?

**Limitations:**

yes

**Strengths And Weaknesses:**

Strengths:
- The paper focuses on an important topic - link stealing attacks for the graph domain data.
- The threat model is correct and reflects real-world scenarios.
- They show experimentally that posterior-based attacks remain effective at extremely low FPR.
- They provide a theoretical analysis of why intra-class links are harder to exploit.
- The new proposed method, PCW, is designed to close the discovered gap.
- The evaluated metrics (AUC, TPR@0.1% - but also include ROC curve) are adequate.
- The paper includes a correct computational complexity analysis.

Weaknesses:
- The proposed PCW method does not significantly outperform the baseline.
- There is not enough information on how this work relates and performs compared to other works in this field.

Minor:
- Table 3. $g$ should be $k$?

---

> ### Author Rebuttal · Authors · 2026-03-31
>
> We thank the reviewer for raising these important points.
>
> **W1&Q1** To directly address this concern, we conducted a paired bootstrap significance analysis comparing PCW against the baseline on the same evaluation pairs using 2,000 paired bootstrap resamples. For each setting, we report the improvement $\Delta = \text{PCW} - \text{Baseline}$, a 95\% bootstrap confidence interval, and a 2-sided paired bootstrap \(p\)-value. An improvement is considered statistically significant if the confidence interval excludes zero and the two-sided  \(p < 0.05\). The results show that PCW does significantly outperform the baseline in several key settings, including citation and molecular graphs. We also include one of the cases we reported in our work where average-case gains do not translate to tail improvements.
>
> ### Table 9: Statistical significance analysis
> | Dataset | Distance | AUC (Baseline) | AUC (PCW) | ΔAUC | 95% CI | p | TPR (Baseline / PCW / Δ / p) |
> |--|--|--|--|--|--|--|--|
> | Cora | Euclidean | 0.818 | **0.855** | **+0.037** | [0.011, 0.065] | **0.009** | 0.120 / **0.287** / **+0.167** / **0.005** |
> |AIDS | Euclidean | 0.874 | **0.918** | **+0.044** | [0.038, 0.050] | **<0.001** | 0.024 / **0.054** / **+0.030** / **0.040** |
> |Photos| Euclidean | 0.896 | **0.907** | **+0.011** | [0.009, 0.013] | **<0.001** | 0.050 / 0.030 / -0.020 / **0.019** |
>
> On Cora with Euclidean distance, PCW yields a statistically significant AUC improvement of +0.037 (\(p=0.009\)) and a statistically significant TPR improvement of +0.167 \(p=0.005\). This corresponds to an increase from 0.120 to 0.287, i.e., more than a 2.3× improvement at the low-FPR tail. Similarly, on AIDS, PCW achieves statistically significant gains in both AUC and low-FPR TPR. At the same time, the significance analysis also includes settings such as Photos, where AUC improves significantly while TPR decreases. We intentionally include this setting to demonstrate that the reported behavior is statistically significant and not attributable to random variation, even when the low-FPR tail does not improve. However, this directly reinforces our central claim that average-case ranking gains do not always translate to improvements at operationally meaningful low-FPR points, which is precisely why our work emphasizes tail-risk evaluation instead of relying solely on AUC.
>
>
> **W2**  Our work is specifically positioned as a revisit of posterior-only black-box link-stealing attacks, with particular focus on the asymmetric vulnerability phenomenon first reported by Zhang et al. [1]. In this sense, the most directly relevant prior work is the original link-stealing framework and its asymmetric attack formulation, which we use as our primary baseline throughout the paper. PCW is not introduced as an unrelated attack, but rather as a principled improvement motivated by our theoretical analysis and empirical diagnosis of posterior-space geometry, specifically the intra-class collapse effect observed under low-FPR evaluation. Accordingly, we rigorously compare PCW extensively against both the original baseline and the improved asymmetric attack setting, across diverse distance metrics, model architectures, and datasets. To further strengthen the empirical comparison, we have now also added results on a larger OGB-scale benchmark, which provides additional evidence regarding scalability and comparative performance (see Table 8 in our reply to Reviewer HSvE).
>
> [1] Zhang et al., *Demystifying Uneven Vulnerability of Link Stealing Attacks against Graph Neural Networks*, ICML 2023.
>
>
> **Q2** A key insight from our attack is that link information is not encoded in individual predictions alone, but in the class-conditional geometry of the posterior space, specifically in the covariance structure exploited by PCW. In particular, training-induced posterior distributions exhibit more structured and anisotropic intra-class geometry, which becomes distinguishable through pairwise comparisons. This suggests that effective defenses should not only regularize individual predictions, but should also control the class-conditional geometry of GNN outputs. One direction is covariance-regularized GNN training, where the model is penalized when the intra-class covariance structure becomes overly informative. Concretely, this may involve constraining the spectrum or encouraging greater isotropy within each class. By limiting highly structured covariance patterns, such a defense would directly suppress the signal exploited by our attack. A second direction is pairwise indistinguishability regularization, which operates directly in the attacker’s space. Since the attack relies on distances between node pairs, the defense can explicitly align the pairwise similarity distributions of training and non-training node pairs, thereby reducing separability in distance space rather than only smoothing individual outputs.
>
> **Minor** It should indeed be $g$. We will update the caption to reflect this correction.

---

> > ### Author Rebuttal · Reviewer_eWSS · 2026-04-02
> >
> > Thank you. I maintain my positive score.

---

> > > ### Author Response · Authors · 2026-04-08
> > >
> > > Thank you for your time and for the helpful review. We truly appreciate your positive assessment of our work.

---

### Official Review · Reviewer_HSvE · 2026-03-13

**Soundness:** 3
**Presentation:** 3
**Significance:** 3
**Originality:** 3
**Overall Recommendation:** 4
**Confidence:** 2

**Summary:**

The paper is interested in the subject of link-stealing attacks in the general context of GNNs, and specifically while considering a black-box setting, where the attacker only have access to posterior predictions from the model. The authors are also interested in the low-false-positive regime, and consider the behavior on the tail-risk, reflecting the ability of the attacker to infer sensitive edges.

**Compliance With Llm Reviewing Policy:**

Affirmed.

**Final Justification:**

I believe that the paper addresses an important privacy problem and makes a compelling case. The proposed PCW method is simple and practical, and the rebuttal addressed my main concerns regarding the theoretical assumptions and some other elements.

At the same time, I also find some of Reviewer yEEz's concerns valid. In particular, the overall novelty of PCW is somewhat moderate, and the practical applicability still depends on posterior access (which was also identified in my own review) and sufficient samples to estimate class-conditional covariance robustly. The experimental setting also remains somewhat limited compared with stronger real-world black-box constraints.

Overall, I think the paper is technically sound, relevant, and provides a useful diagnostic and evaluation perspective. Please note again that while I work in GNNs, I don't work in this specific direction, and therefore take into account my low confidence here. So overall the manuscript's contribution may be seen as incremental than groundbreaking, and some limitations remain unsolved by the authors even tough I believe the rebuttal clarified some elements and provided some new empirical evidence. I would maintain my "weak accept" recommendation and would rely on the other reviewers and the AC to judge if these elements are enough to reach the acceptance bar of ICML. In any case, I believe the authors should continue enhancing the manuscript.

**Key Questions For Authors:**

- Could you clarify some assumptions in the theoretical analysis?
- Could you provide some elements regarding the empirical analysis of time complexity?
- Any insights why PubMed as a dataset shows weaker improvement than other datasets ? Is it related to posterior concentration effect?
- Could you provide results on rather newer datasets like the OGB ? This is mainly to understand the scalability of the method.

**Limitations:**

- The main bottleneck of the attack is the required access to the posterior probability vectors. I don’t know if that counts within the black-box setting (specifically in cases the attacker have only access to the final predictions).
- There are questions regarding the scalability of the issue and the number of candidates than can be expensive for large graphs
- The results empirically varies between datasets, questioning therefore the generality and effectiveness.

Please note that I am not very knowledgeable in this space, which is reflected by my confidence score.

**Strengths And Weaknesses:**

**Strengths:**
- The paper is interested in the problem of link privacy leakage, which is a very important subject and very relevant in the case of GNNs deployed in different domains (for instance like healthcare graphs).
- The interest shift from the classical average-case (through AUC for instance) to rather tail-risk is well motivated and makes total sense. In practical setting, we could expect that the attacker care about a small number of high-confidence success.
- The theoretical analysis is very interesting where the authors shows the intra-class posteriors share the same dominant coordinate, showing therefore that intra-class score distributions can overlap closely.
- The proposed Per-Class Whitening (PCW) method is practical and simple.


**Weaknesses:**
- There is a clear lack of defense analysis on the empirical side. Specifically, the paper didn’t explore any defense mechanisms or mitigation strategies, limiting therefore the practical impact.
- Regarding the theoretical analysis, the authors assumes a certain independence of posterior coordinates, which may be limited in real practice since the correlation exists between classes, and also the posterior distributions may be highly structured (given that they come from a softmax layer). It would be great if the authors could clarify these elements and assumptions.
- For the PCW attack, the matrix-vector multiplication can be a little bit complex in terms of operation complexity and therefore the complexity (as denoted in Section G). Did you provide some specific time complexity in this case, where we can see how it does in empirical setting?

---

> ### Author Rebuttal · Authors · 2026-03-31
>
> We thank the reviewer for the insightful comments.
>
> **W1** Please see our replies to Reviewers yEEz and eWSS on defense evaluation and discussion.
>
> **Q1** We agree that posterior vectors produced by a Softmax layer are inherently correlated due to the simplex constraint and shared normalization, and thus do not satisfy independence in practice. Our theoretical analysis does not assume independence as a requirement of the method. Rather, the simplified decomposition is introduced solely for analytical tractability, allowing us to isolate how scaling in posterior space affects distance behavior and separability. Our proposed method does not rely on this simplification. On the contrary, it explicitly models the structured dependencies between posterior coordinates through class-conditional covariance estimation. These correlations are central to the method’s effectiveness, as they capture the anisotropic structure of posterior space that standard distance metrics fail to exploit. From our experiments, variants that ignore cross-coordinate dependencies (e.g., diag norm, as shown in Tables 1& 6 in our response to Reviewer yEEz) underperform compared to full covariance-based PCW. This demonstrates that capturing these structured correlations is essential for improving intra-class separability.
>
> **Q2&L2** The table below reports runtime of the baseline Euclidean attack and PCW. PCW is decomposed into a one-time fit and a per-pair attack cost.
>
> ### Table 7: Time cost
> Dataset|#Pairs|Baseline(s)|Fit(s)|Attack(s)|PCW Total(s)
> --|--|--|--|--|--
> Cora|1,056|0.032|0.034|0.035|0.069
> Citeseer|912|0.028|0.030|0.031|0.061
> PubMed|8,866|0.265|0.268|0.410|0.678
> AIDS|6,478|0.199|0.213|0.238|0.451
> |Photos|23,818|0.745|0.742|0.829|1.571|
> |CS|16,380|0.492|0.537|0.547|1.085|
> |OGBN-Products|12,371,804|399.571|465.576|495.305|960.881
>
> From an attacker’s perspective, PCW is practical. The fit stage is computed once and reused, while the attack stage scales linearly with the number of pairs. Even at large scale, PCW remains efficient. On OGBN-Products (12.3M pairs), it completes in 961 seconds and &asymp;40 μs per pair post-fitting, enabling attacks over millions of samples within minutes in a black-box setting.
>
> **Q3&L3** The weaker improvement on PubMed is due to posterior concentration effects, as analyzed in Appendix F. PubMed exhibits lower mean effective posterior support, indicating more peaked posteriors and reduced variability, which limits the structure PCW can exploit. Additionally, PubMed has a highly concentrated negative tail, where a small set of tightly clustered negatives determines the low-FPR threshold. In this regime, improvements in global separability may not translate to higher TPR at extremely low FPR. More broadly, performance variation reflects differences in posterior geometry. PCW consistently improves performance, while dataset-specific geometry governs gains at strict operating points.
>
> **L1** Our attack operates in the standard score-based black-box setting, where the adversary can query the model to obtain posterior probabilities (confidence scores) without access to model parameters or training data. This setting is widely adopted in prior work on membership inference and link-stealing attacks and reflects common MLaaS APIs, such as in AWS SageMaker. In the strictest label-only setting (top-1 prediction only), posteriors collapse to one-hot vectors, eliminating intra-class variability. To bridge these settings, we also evaluate limited-output regimes (Top-2/Top-3) in Table 4 (response to Reviewer yEEz). PCW remains effective and consistently outperforms baselines, demonstrating robustness under partial posterior access.
>
> **Q4** We evaluate our method on the large-scale ogbn-products dataset.
>
> ### Table 8: Result showing AUC/TPR@0.001/TPR@0.01
> Model|Setting|Cos|Eucl|SqEuc|Corr|City|Cheb|Bray|Canb
> --|--|--|--|--|--|--|--|--|--
> GCN|Intra|0.573/0.027/0.059|0.517/0.000/0.003|0.517/0.000/0.003|0.593/0.106/0.179|0.542/0.019/0.041|0.562/0.000/0.003|0.542/0.019/0.041|0.573/0.000/0.009
> ||PCW|**0.777/0.161/0.302**|**0.682/0.081/0.175**|**0.682/0.081/0.175**|**0.784/0.160/0.301**|**0.667/0.068/0.150**|**0.682/0.083/0.180**|**0.756/0.117/0.252**|**0.713/0.073/0.159**
> GAT|Intra|0.613/0.002/0.015|0.631/0.002/0.015|0.631/0.002/0.015|0.611/0.002/0.015|0.637/0.002/0.015|0.628/0.002/0.015|0.637/0.002/0.015|0.645/**0.022/0.072**
> ||PCW|**0.766/0.003/0.028**|**0.695/0.002/0.021**|**0.695/0.002/0.021**|**0.771/0.003/0.031**|**0.692/0.002/0.021**|**0.684/0.002/0.021**|**0.750/0.003/0.026**|**0.731**/0.004/0.041
> SAGE|Intra|**0.650**/0.006/0.047|0.619/0.006/0.059|0.619/0.006/0.059|0.645/0.006/0.047|0.638/0.006/0.056|**0.610**/0.006/0.054|0.638/0.006/0.056|0.547/**0.018**/0.044
> ||PCW|0.646/**0.007/0.089**|**0.634/0.007/0.086**|**0.634/0.007/0.086**|**0.645/0.007/0.092**|**0.644/0.007/0.079**|0.606/**0.007/0.080**|**0.640/0.007/0.087**|**0.619**/0.006/**0.077**
>
> The results demonstrate that PCW scales to large & realistic settings.

---

> > ### Author Rebuttal · Reviewer_HSvE · 2026-04-02
> >
> > I thank the authors for the additional clarifications. I believe that I understand better the proposed method and overall contribution now, and I maintain my already positive score while keep monitoring the other reviews to see if I have missed something.

---

> > > ### Author Response · Authors · 2026-04-08
> > >
> > > Thank you for your thoughtful review and for taking the time to read our rebuttal. We sincerely appreciate your positive assessment and are glad that the additional clarifications helped better convey the proposed method and its contribution.

---

### Official Review · Reviewer_yEEz · 2026-03-13

**Soundness:** 2
**Presentation:** 3
**Significance:** 3
**Originality:** 2
**Overall Recommendation:** 3
**Confidence:** 4

**Summary:**

This paper revisits posterior‑only, black‑box link‑stealing attacks on GNNs and argues that privacy should be evaluated as a tail‑risk problem (e.g., TPR at very low FPR) rather than with average‑case AUC. It attributes the inter‑ vs. intra‑class vulnerability gap to a geometric bottleneck in posterior space: for intra‑class pairs, aligned dominant coordinates collapse distances and suppress discriminative signal. The paper proposes Per‑Class Whitening (PCW) in a power‑transformed posterior space to re‑condition covariance per predicted class before computing distances.

**Compliance With Llm Reviewing Policy:**

Affirmed.

**Key Questions For Authors:**

1. How does PCW differ conceptually and empirically from class‑conditional Mahalanobis scoring applied to logits or calibrated features? Please include direct baselines comparisons and, if possible, theoretical reasoning showing PCW’s advantage beyond a re‑scaled Euclidean metric.

2. For cases where AUC/TPR = 1.0, can you rule out leakage or label‑based grouping on the test set? What is the protocol for threshold selection and how is it separated from test evaluation? Please re‑report with mean and std across multiple random splits.

3. Under a strict black‑box API, how does the adversary obtain enough posterior samples per class to estimate covariances robustly, especially with rare classes? What is the sample complexity, and how does PCW behave when posteriors are truncated/rounded (e.g., top‑k only) which is common in practice?

4. Can you evaluate PCW on OGB datasets and inductive GNNs, and under label‑only or noisy posterior interfaces, to reflect current MLaaS practices? Also, how does PCW fare against defenses (e.g., differential privacy (DP), temperature scaling, posterior noise) that are known to affect membership/link inference?

5. Can the authors please provide ablations on the power transform $\tau$, covariance regularization, number of queries per class, calibration (temperature scaling), and class imbalance. Which component of PCW is most responsible for the reported gains, and do those gains persist when evaluated across many seeds/splits?

**Limitations:**

Yes

**Strengths And Weaknesses:**

Strengths:

- Measuring TPR at very low FPR aligns with the privacy literature (e.g., LiRA’s critique of average‑case metrics), and bringing that emphasis to link stealing is a reasonable contribution to evaluation practice.
- The paper’s diagnosis of  inter‑ vs. intra‑class asymmetry is consistent with prior findings on uneven vulnerability in link‑stealing MIAs[1].
- Simple, reproducible attack tweak. PCW is easy to implement (class‑conditional whitening in a transformed posterior space), making the empirical comparison straightforward.


Weaknesses:
- PCW is essentially class‑conditional whitening/Mahalanobis‑style re‑scaling applied to posteriors; similar ideas are standard in OOD/confidence estimation [2], and the paper lacks stronger theoretical justification that PCW improves Bayes error for intra‑class mixtures.
-  Some results indicate perfect separation (AUC/TPR = 1.0) for certain Citeseer splits, which is suspicious and may suggest leakage or protocol issues.
-  Experiments largely use small transductive citation graphs that are known to produce misleading rankings; the work does not evaluate on large-scale datasets e.g. OGB graphs with realistic splits, inductive settings, or restricted APIs (label‑only/rounded posteriors) that better reflect practice. Additionally, many services expose only labels or rounded top‑k posteriors; evaluating PCW under these constraints would make the threat model credible.
- It is unclear how an attacker robustly estimates per‑class covariance from black‑box queries at scale, or how PCW behaves under posterior truncation/noise or calibration shifts.

---

> ### Author Rebuttal · Authors · 2026-03-31
>
> We thank the reviewer for the helpful feedback.
>
> **Q1** PCW differs from class-conditional Mahalanobis in both objective and geometry. Mahalanobis is a pointwise centroid-based score, whereas PCW is a pairwise link-inference metric that whitens posterior differences. It is also different to scalar Euclidean rescaling, since the posterior covariance is highly anisotropic, causing different eigendirections to be weighted differently. As shown in Table 1 (Cora + GAT + Euclidean), PCW improves over simpler rescaling baselines, indicating that the gain comes from full class-conditional covariance reconditioning.
>
> ### Table 1: Simple rescaling baselines vs PCW.
> |Method|AUC|TPR@0.1%|
> |--|--|--|
> |Raw|0.818|0.086|
> |$\tau$-only|0.858|0.244|
> |Class-centered|0.858|0.244|
> |Diagonal whitening|0.849|0.222|
> |Global whitening|0.841|0.147|
> |PCW|**0.874**|**0.294**|
>
> **Q2** We rule out leakage or label-based grouping, as all groups are defined from model posteriors. Thresholds are selected on calibration data and fixed before test evaluation, then applied once to the held-out test set. We have performed a label-permutation check (Table 2), yielding chance-level performance with permutation, and confirming no leakage. Also, AUC/TPR = 1.0 in the CC-LS experiment arises from Citeseer sparsity; in the $g_1$ subgroup, some splits contain only 2 positive pairs, making estimates unstable at the tail for Citeseer as shown in Table 2.
>
> ### Table 2: Inter–CC-LS ($g_1$) for cosine and correlation distance metrics.
> | Run | Dist | AUC | TPR|
> |--|--|--|--|
> |Permuted|Cosine|0.559 ± 0.047|0.022 ± 0.027|
> | |Correlation |0.548 ± 0.034|0.026 ± 0.024|
> |Actual| Cosine|0.836 ± 0.124|0.125 ± 0.354|
> | |Correlation| 0.921 ± 0.067|0.167 ± 0.356|
>
>
> **Q3** PCW does not require uniform class coverage, as posteriors are concentrated in a low-dimensional effective space. Rare classes have fewer samples, but this concentration enables reliable covariance estimation even with modest counts. Also, we do not assume a closed-form sample complexity; instead, we evaluate empirically. With reduced pair fractions (10–50%), PCW remains stable and consistently improves over the baseline, indicating robustness under limited samples.
>
> ### Table 3: Sample efficiency (Cora+cosine):
> |Fraction|Intra TPR|PCW TPR|
> |--|--|--|
> |0.1|0.133|0.489|
> |0.3|0.353|0.500|
> |0.5|0.191|0.479|
>
> Under truncation, PCW remains effective for top-2/3 but degrades to chance for top-1, confirming reliance on posterior structure.
>
> ### Table 4: Top-k truncation (Cora+cosine):
> |Setting|Intra TPR|PCW TPR|
> |--|--|--|
> |Top-1|0.001| 0.001|
> |Top-2| 0.043| 0.045|
> |Top-3| 0.039| 0.054|
>
> **Q4** Please see the results for the large-scale ogbn-products benchmark in the response to Reviewer HSvE, where results demonstrate scalability beyond citation graphs. We study posterior-only access in transductive GNN APIs, consistent with prior link-stealing work, where attacks are typically first established in the transductive setting before being extended to inductive variants. Our goal is to characterize worst-case risk under posterior-only access, which we evaluate extensively in the transductive regime. Under defenses, we evaluate PCW across increasing perturbation strength. Performance degrades as noise increases, while remaining strong under moderate perturbations, indicating robustness to realistic noise levels. PCW consistently outperforms the baseline across these settings, especially at the tail.
>
> ### Table 5: Representative results due to space (Citeseer+GAT+Cosine):
> |Defense|Value|AUC|TPR|PCW AUC|PCW TPR|
> |--|--|--|--|--|--|
> |None|0| 0.907| 0.255 |**0.937**|**0.466**|
> |Temp|0.7| 0.885| 0.213|**0.933**|**0.353**|
> |Temp|1.5| 0.921| 0.317|**0.942**|**0.518**|
> |Gaus|0.05| 0.901| 0.120|**0.909**|**0.159**|
> |Gaus|0.1| 0.871| 0.042|**0.849**|**0.235**|
> |Lap|0.05| 0.892| 0.119|**0.901**|**0.241**|
> |Lap|0.1| 0.859| 0.050|**0.844**| **0.273**|
> |DP|10| **0.859**| 0.050|0.844| **0.273**|
> |DP|5| **0.773**| 0.020|0.729| **0.038**|
>
> **Q5** We ablate the PCW components (Table 6), varying one factor at a time. Results over $\tau$ (0.841–0.872) and covariance (diag 0.812 vs. Ledoit–Wolf 0.855) show gains are driven by posterior transformation and covariance modeling. Query budget has limited impact, class imbalance is negligible, and temperature scaling is slightly beneficial.
>
> ### Table 6: Ablation (citeseer+GAT+Cosine)
> |Comp|Setting|AUC|TPR|
> |--|--|--|--|
> |Base|--|0.803|0.195|
> |$\tau$|0.25|0.857|0.328|
> |$\tau$|0.5|0.849|0.320|
> |$\tau$|0.75|0.841|0.312|
> |$\tau$|1|0.872|0.300|
> |Query budget (Q)| 32| 0.830 | 0.285 |
> |Q|64|0.843| 0.292 |
> |Q|128| 0.848 | 0.320 |
> | Q | 256 | 0.848 | 0.314 |
> | Regularization mode (C) | Diag. | 0.812 | 0.270 |
> | C | None | 0.849 | 0.320 |
> | C | Ledoit | 0.855|0.328|
> | C | Floor | 0.849|0.321|
> | Temp (T) | 1 | 0.849 | 0.320 |
> | T| 1.5 | 0.860 | 0.326 |
> |T|2|0.865| 0.330 |
> |T|3| 0.871|0.333|
> |Imbalance Ratio (I)|0.25|0.839|0.318|
> |I| 0.5 | 0.845 |0.325|
> |I| 0.75 | 0.846|0.314|
> |I|1.0|0.849|0.320|

---

> > ### Author Rebuttal · Reviewer_yEEz · 2026-03-31
> >
> > Thanks for the rebuttal, I am keeping my score

---

> > > ### Author Response · Authors · 2026-04-08
> > >
> > > Thank you for taking the time to read our rebuttal and for the helpful feedback throughout the review process. We appreciate your acknowledgement that the concerns were addressed.

---

### Decision · Program_Chairs · 2026-04-30

**Decision:**

Accept (regular)

**Comment:**

The paper frames posterior-only black-box stealing attacks as a tail risk problem and argues that the performance should be evaluated using TPR at low FPR metrics. The reviewers praised the framing of the problem and found the proposed "Per-Class Whitening (PCW) method" to be practical and simple. The proposed method was inspired by an insightful theoretical analysis of the geometry of the posterior distribution.

Reviewer eWSS had concerns that PCW does not outperform the baseline, and pointed out a lack of comparison with other works. In reply, the authors conducted a paired bootstrap significance analysis and showed statistical significance, and clarified the positioning of their work in the context of existing literature. This reviewer maintains their positive score.

Several reviewers had questions about defenses. The authors suggested covariance-regularized GNN training and pairwise indistinguishability regularization as two promising directions for developing a defense, but leave this for future work. The also evaluate PCW across increasing perturbation strength and temperature scaling as simple "sanity check" defeneses, with predictable results.

Reviewer HSvE had questions about the assumptions behind the theory and the runtime. The authors sufficiently addressed these concerns. Their method remains practical even at large scales (e.g. OGBN-Products).

Reviewer yEEz was the most critical and requested several additional experiments. In response the authors added rescaling baselines, results on sample efficiency, top-k truncation, larger scale datasets, simple defenses, as well as some ablation studies. They also provided some evidence to rule out leakage since the reviewer was suspicious of some results. Overall, even though the reviewer maintains a negative score they picked the "Fully resolved" option in the acknowledgment. in my opinion the authors have done a good job addressing the concerns.

Given the above I recommend acceptance.